# Proteomic profiling and genome-wide mapping of O-GlcNAc chromatin-associated proteins reveal an O-GlcNAc-regulated genotoxic stress response

Yubo Liu [1,3], Qiushi Chen [2,3], Nana Zhang[1], Keren Zhang[2], Tongyi Dou[1], Yu Cao[1], Yimin Liu[1], Kun Li[1], Xinya Hao[1], Xueqin Xie[1], Wenli Li[1], Yan Ren [2✉] & Jianing Zhang [1✉]

O-GlcNAc modification plays critical roles in regulating the stress response program and cellular homeostasis. However, systematic and multi-omics studies on the O-GlcNAc regulated mechanism have been limited. Here, comprehensive data are obtained by a chemical reporter-based method to survey O-GlcNAc function in human breast cancer cells stimulated with the genotoxic agent adriamycin. We identify 875 genotoxic stress-induced O-GlcNAc chromatin-associated proteins (OCPs), including 88 O-GlcNAc chromatin-associated transcription factors and cofactors (OCTFs), subsequently map their genomic loci, and construct a comprehensive transcriptional reprogramming network. Notably, genotoxicity-induced O-GlcNAc enhances the genome-wide interactions of OCPs with chromatin. The dynamic binding switch of hundreds of OCPs from enhancers to promoters is identified as a crucial feature in the specific transcriptional activation of genes involved in the adaptation of cancer cells to genotoxic stress. The OCTF nuclear respiratory factor 1 (NRF1) is found to be a key response regulator in O-GlcNAc-modulated cellular homeostasis. These results provide a valuable clue suggesting that OCPs act as stress sensors by regulating the expression of various genes to protect cancer cells from genotoxic stress.

---

[1] School of Life and Pharmaceutical Sciences, Dalian University of Technology, Panjin, China. [2] Clinical Laboratory of BGI Health, BGI-Shenzhen, Shenzhen, China. [3] These authors contributed equally: Yubo Liu, Qiushi Chen. ✉email: reny@bgi.com; jnzhang@dlut.edu.cn

Dynamic and reversible modification of nuclear and cytoplasmic proteins with an O-linked β-N-acetylglucosamine (O-GlcNAc) monosaccharide have been implicated in diverse cellular processes, including gene expression, signal transduction, the cell cycle, and metabolism[1]. Unlike canonical prototypical glycosylation, O-GlcNAc is not further extended to oligosaccharides. Only two enzymes, O-GlcNAc transferase (OGT) and O-GlcNAcase (OGA), are responsible for the introduction and removal of O-GlcNAc, respectively[2]. Notably, the intracellular O-GlcNAc modification level has been found to respond to extracellular signals and stimuli, such as the glucose concentration[3], hormones[4] and cellular stress[5,6].

In response to hostile intracellular and extracellular stresses such as hypoxia, nutrient deprivation, and genotoxic stimuli, cells usually utilize specialized pathways to counteract deleterious effects and maintain homeostasis, and these pathways are termed the cellular stress response[7]. Emerging evidence has implicated that O-GlcNAc modification in the cellular stress response[5,6,8–10]. Other research groups and ours have observed that cancer cells responded to stimuli by elevating their O-GlcNAc levels in a stress-, time- and dose-dependent manner through an increased synthesis of the sugar donor UDP-GlcNAc[11–14]. Alterations in the intracellular O-GlcNAc modification likely affect various key proteins, these effects might protect cells against genotoxic stress and ultimately induce drug resistance in cancer cells[6,15,16].

One of the possible mechanisms involved in the O-GlcNAc-regulated stress response might be the modulation of gene transcription[17,18]. Chromatin-associated proteins, including transcription factors (TFs) and their cofactors, play essential roles in the conversion of stress signals to perceptive transcriptional reprogramming by binding to the promoter region of various target genes and thus ensuring the rapid activation of the necessary adaptive signaling cascades[19]. Recently studies discovered that chromatin O-GlcNAc modification is involved in gene expression[20,21]. Abundant TFs and cofactors involved in the transcriptional regulatory machinery have been found to carry O-GlcNAc modifications, which influence their stability, transcriptional activity, nuclear localization, and protein–protein and protein-DNA interactions[17,22]. Diverse insults, including genotoxicity, markedly change the O-GlcNAc modification of numerous TFs, as observed in both in vitro and in vivo studies. The genotoxic agent adriamycin (Adm) reportedly induces O-GlcNAc modification of the tumor suppressor p53 and thereby, dynamically regulates its stability and activity[15]. Therefore O-GlcNAc modification of TFs may serve as a stress sensor and regulate the expression of stress-response genes. Chromatin immunoprecipitation (IP) followed by next-generation sequencing (ChIP-seq) has enabled the genome-wide profiling of the DNA-binding sites of individual TFs[23]. The Vocadlo group reported a chemical reporter-based (per-O-acetyl N-azidoacetyl galactosamine, Ac₄GalNAz) ChIP-seq-like approach for the genome-wide mapping of the DNA-binding sites of metabolically labeled O-GlcNAc proteins in Drosophila[24]. Using this chemical-genetic method in a time-course study, these researchers monitored the turnover of O-GlcNAc on chromatin and determined that genomic loci exhibit varying O-GlcNAc behavior[25]. However, studying the mechanism through which a stimulus orchestrates the O-GlcNAc modification of multiple TFs and subsequently modulates transcriptional networks remains a challenge.

Herein, we adapt the above-mentioned chemical reporter approach to develop a multiomics strategy for the proteomic profiling and genome-wide mapping of genotoxic stress-induced O-GlcNAc chromatin-associated proteins (OCPs) in human breast cancer cells. The metabolically labeled OCP subproteome is quantitatively profiled to characterize the O-GlcNAc chromatin-bound TFs and cofactors (OCTFs) that show activity under Adm stimulation. The enriched O-GlcNAc-bound DNA is implemented to unambiguously discover target genes of OCTFs, examine the genome-wide stimulus-influenced dynamics of TF binding, and ultimately link these findings to subsequent transcriptome changes. We provide evidence showing that dynamic changes in OCP loci in the genome determine the differential gene expression patterns and cellular survival under genotoxic conditions. This multiomics method will also have broad applications for studying O-GlcNAc-regulated gene expression in other cell or tissue models and potentially characterizing functionally unknown genes in O-GlcNAc-associated pathways.

## Results

**Global OCPs are biased to transcriptionally participate in the stress response.** The chemical reporter strategy for metabolic glycan labeling exploits unnatural monosaccharides bearing a bioorthogonal functional group (e.g., an azide), which can be metabolically incorporated into cellular glycans. And this incorporatin allows subsequent conjugation with imaging probes or enrichment tags via click chemistry[26]. For the labeling of O-GlcNAc, we employed N-azidoacetylgalactosamine (GalNAz) in its nonacetylated form to avoid the recently discovered nonspecific reaction with cysteine induced by per-O-acetylated GalNAz (Ac₄GalNAz)[27]. Two breast cancer cell lines, MCF-7 and MDA-MB-231 cells, were incubated with 1 mM GalNAz for 48 h, and the proteins were the crosslinked with chromatin using 1% formaldehyde. The crosslinked chromatin was isolated, fragmented and reacted with alkyne-biotin. Formaldehyde crosslinking captures the protein–protein interactions (PPIs) between OCPs and other secondary or remote proteins. To eliminate the risk of nonspecific contaminations, 2% SDS[28] was used to effectively reverse the formaldehyde crosslinks. OCPs were then enriched with streptavidin-beads and strictly washed with standard low-salt and LiCl buffer for ChIP (Supplementary Fig. 1a). The resulting proteins were subjected to immunoblotting or liquid chromatography-tandem mass spectrometry (LC-MS/MS) based proteomic analysis (Fig. 1a).

Immunoblotting of cell lysates and chromatin precipitates from GalNAz-treated cells showed the incorporation of azides into O-GlcNAc proteins (Fig. 1b). We subsequently analyzed the OCPs in MCF-7 cells by LC-MS/MS. A total of 990 azide-labeled O-GlcNAc (O-GlcNAz) proteins were identified with high confidence (identified at least six times in nine biological replicates, Supplementary Fig. 1b–c and Supplementary Data 1). To further reduce the influence of the contamination caused by nonspecific interactions during the procedure used for O-GlcNAz protein capture, only the candidate proteins located in the nucleus were regarded as the OCPs. Ontological analyses of 575 filtered OCPs revealed enrichment of terms related to RNA splicing and ribosome regulation[29,30], consistent with previously known functions, as well as the additional terms cellular stress response, transcriptional regulation and chromatin remodeling (Fig. 1c). These data imply that O-GlcNAc-associated transcriptional regulation might interfere with the cancer cell stress response, even in the absence of stress conditions.

To further verify this hypothesis, the putative functional processes associated with 166 OCTFs among the identified MCF-7 OCPs were annotated. A network of these specific functional complexes was constructed, and a PPI analysis showed that the cellular stress response group shared nodes with the chromatin remodeling and transcriptional regulation groups, which suggests that the OCTFs are related to stress-response gene transcription (Fig. 1d and Supplementary Data 2). Further assays verified that multiple OCTFs and stress-responsive gene promoter regions were

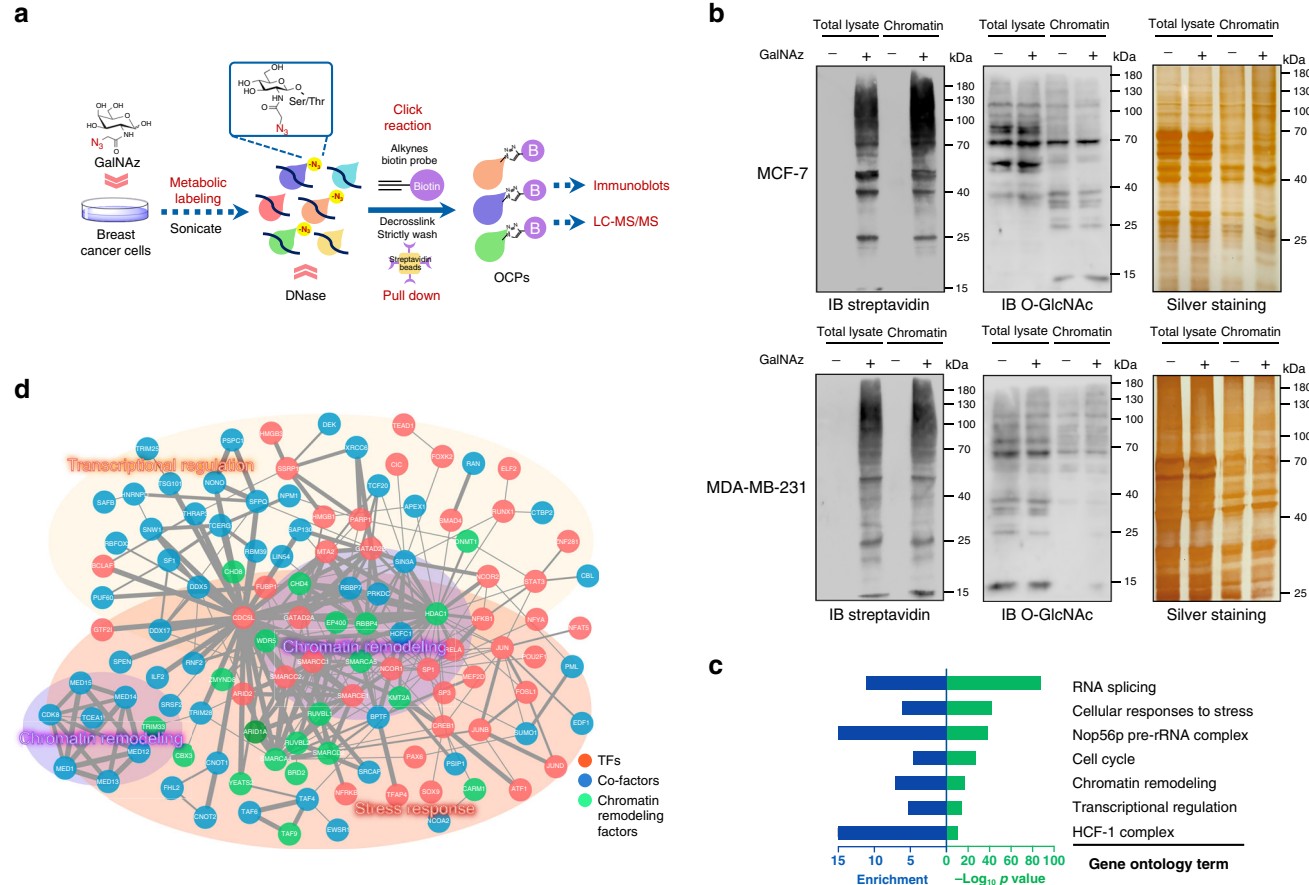

**Fig. 1 OCPs are related to the stress response and transcriptional regulation. a** Chemical reporter-based metabolic labeling method used to analyze the OCPs by immunoblotting and LC-MS/MS. MCF-7 and MDA-MB-231 cells were incubated in 1 mM GalNAz media or vehicle for 24 h. Azide (N$_3$) is shown in bold in the chemical structure. Biotin-alkynes (B) was used. Chromatin was crosslinked and conjugated to alkyne-biotin by a downstream click reaction. O-GlcNAz chromatin was subsequently decrosslinked and enriched with streptavidin-beads for immunoblotting and LC-MS/MS. **b** Immunoblotting and silver staining of GalNAz labeled MCF-7 and MDA-MB-231 cell lysates and chromatin. Chromatin proteins of cancer cells in the presence and absence (vehicle) of GalNAz (24 h) were extracted and reacted with alkyne-biotin, and subjected to immunoblotting with streptavidin and the O-GlcNAc-specific antibody CTD110.6. All blots and sliver staining are representative of at least two biologically independent experiments. **c** GO terms enriched for OCPs in MCF-7 cells. **d** STRING PPI analysis and functional enrichment analysis of OCTFs in MCF-7 cells. **b**, **c** Source Data are provided as a Source Data file.

enriched in the O-GlcNAz chromatin precipitate (Supplementary Fig. 1b, d). These results suggest that OCTFs are likely to participate to the cellular stress response through gene transcription.

**Genotoxic stress-induced O-GlcNAc enhances the genome-wide interactions of OCPs with chromatin**. To intensively explore the cellular stress sensor function of O-GlcNAc, a genotoxicity-adapted cell model was developed. The parental MCF-7 cells were exposed to stepwise increasing concentrations of the genotoxic stress-inducing agent Adm[31,32] over an 8-month period. The IC$_{50}$ of the Adm-resistant variant (MCF-7/ADR, ADR) was increased by ~30-fold (0.32–10.1 μM, Supplementary Fig. 2). Compared with the parental MCF-7 cells, the ADR cells displayed marked striking increases in whole-cell lysates and chromatin-bound O-GlcNAc (Fig. 2a and Supplementary Fig. 3). A similar tendency was also found with other genotoxicity-adapted cell models (Supplementary Fig. 4). Gene expression profiling by RNA-seq identified 7112 differentially expressed genes (DEGs, false discovery rate (FDR) ≤ 0.001 and fold change ≥ 2) between ADR and MCF-7 cells (Fig. 2b and Supplementary Data 3). Similar to the Gene Ontology (GO) enrichment analysis (Supplementary Fig. 5), a gene set enrichment analysis (GSEA) revealed that ADR cells express genes involved in cellular stress and the DNA damage response at

higher levels than MCF-7 cells (Fig. 2c), which indicates that cancer cells adjust to genotoxic stress through chromatin O-GlcNAc fluctuations and the transcriptional regulation of many genes.

For the genome-wide assessment of O-GlcNAc-regulated genes in the above-described cell models, we designed an integrative omics strategy that combines three global datasets: chemical reporter-based OCP quantitative proteomics, chemoselective O-GlcNAc chromatin loci (chemoselective O-GlcNAc chromatin sequencing, COGC-seq) and O-GlcNAc-regulated transcriptomics. The workflow of this strategy is illustrated in Fig. 2d. An integrative analysis of differential quantitative OCPs from proteomics datasets, and enriched TF motifs from COGC-seq datasets allowed us to identify genotoxic stress-induced OCTFs. Furthermore, the differentially expressed COGC-seq peak-associated genes were matched to genotoxicity-induced transcriptome. After systematical study, a genome-wide mechanism of O-GlcNAc regulated gene expression emerged.

Quantitative proteomics comparisons between nine biological replicates of MCF-7 and ADR O-GlcNAc chromatin that exhibited high correlations identified a total 1952 proteins (Supplementary Fig. 6a–c), and after valid value filtering (a protein must be identified in six out of nine replicates of at least one group), we pursued the further analysis of 1403 proteins.

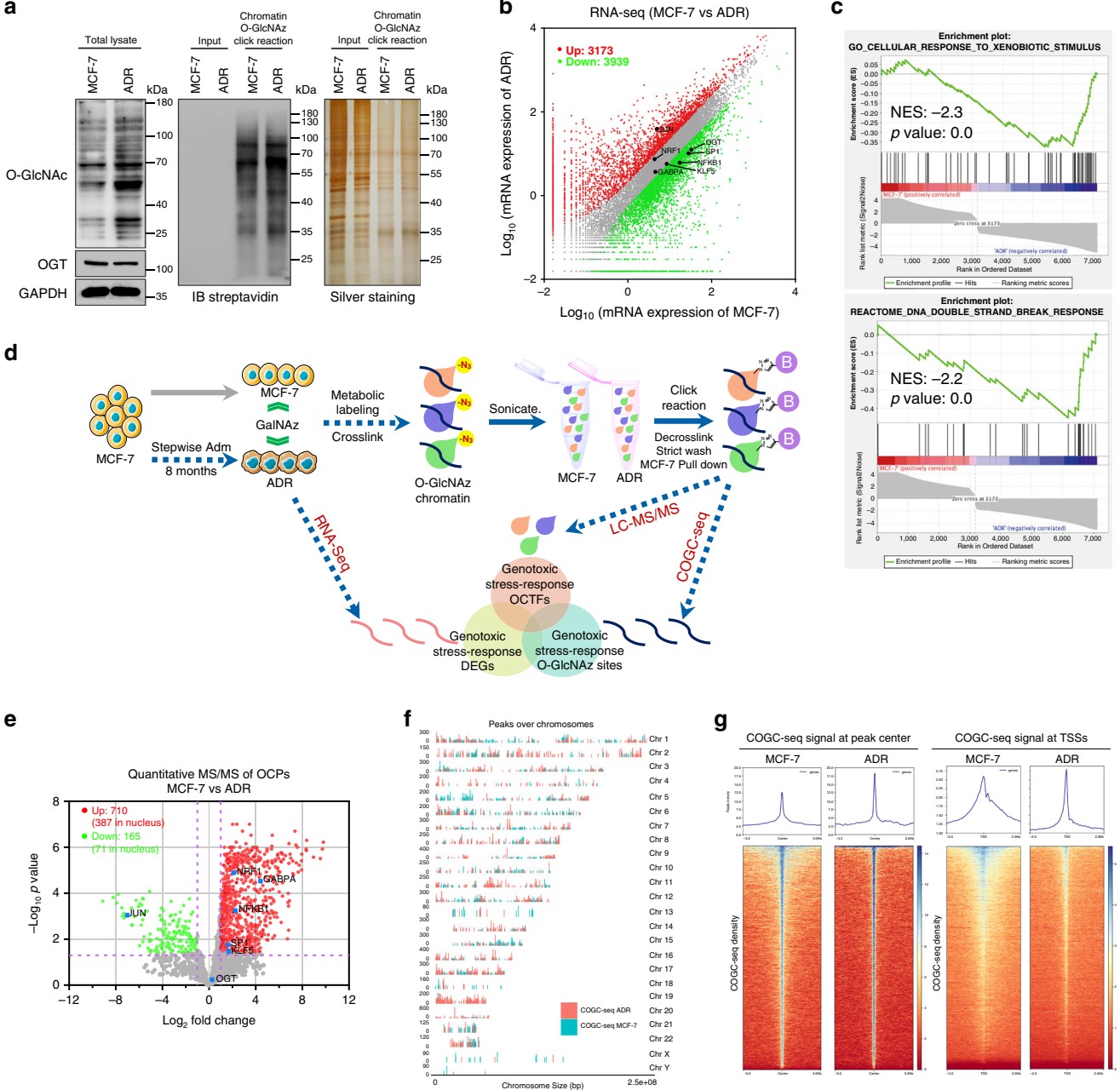

**Fig. 2 Genome-wide multiomics analysis of OCPs in genotoxic stress-influenced cell models. a** Immunoblotting and silver staining of GalNAz labeled MCF-7 and ADR cell lysates and chromatin. Chromatin proteins of the cells in the presence of 1 mM GalNAz (24 h) were extracted and reacted with alkyne-biotin. The cellular O-GlcNAc and expression of OGT were analyzed by immunoblotting. All blots and sliver staining are representative of at least two biologically independent experiments. Source Data are provided as a Source Data file. **b** Scatter plot showing the RNA-seq DEG expression levels (fragments per kilobase of transcript per million mapped reads (FPKM), fold change ≥ 2, FDR ≤ 0.001) in MCF-7 vs. ADR cells. Genes mentioned in this study are labeled. $n = 2$ biologically independent RNA-seq replicates. **c** GSEA of RNA-seq DEGs in MCF-7 cells compared with ADR cells. The normalized enrichment score (NES) and $p$ value are indicated. **d** Schematic of the multiomics strategy involving a genome-wide chemical reporter-based method. MCF-7 cells were exposed to stepwise increasing concentrations of Adm for 8 months to generate the genotoxic resistant variant ADR cells. MCF-7 and ADR cells were incubated on 1 mM GalNAz media for 24 h. Decrosslinked OCPs were compared by label-free relative quantitative proteomics. The genomic DNA fragments bounded by OCPs were decrosslinked and then subjected to next-generation sequencing (COGC-seq). DEGs between MCF-7 and ADR cells were identified by RNA-seq analysis. An overlap analysis of three layered omics datasets was subsequently performed to uncover the O-GlcNAc regulated genotoxic stress-responsive reprograming. **e** Volcano plot of label-free relative quantitative proteomics data of OCPs in MCF-7 and ADR cells ($n = 9$ biologically independent experiments, two-sided unpaired Student's $t$-test). The OCPs mentioned in this study are labeled. **f** Coverage plot of the COGC-seq peak locations over the whole human genome. The peaks of ADR cells showed distinct characteristic with those of MCF-7 cells. **g** Heat maps of the COGC-seq signal density at the peak center and TSSs (±3 kb). The average signal profile is shown. The red color indicates low a signal, and a blue color indicates a high signal.

Subsequent statistical analyses showed that 875 OCPs (458 annotated in the nucleus), including 88 TFs and cofactors, exhibited ≥2-fold differences ($p$ value ≤ 0.05 and FDR ≤ 0.05, Fig. 2e, Supplementary Data 4 and 5). Most OCPs increased their interactions with chromatin after genotoxic adaptation. The vast majority of OCP quantities could not reflect the gene expression differences in the transcriptome and whole-cell proteomics without O-GlcNAz enrichment, which indicates the specificity of this chemical reporter-based enrichment (Supplementary Figs. 6d–f, 7a, b, Data 6). Correspondingly, the GSEA and GO analysis showed that the OCPs were predominantly involved in the cellular stress response and other genotoxic stress-related processes (DNA repair, cell cycle and apoptosis, Supplementary Fig. 7c, d). Invisible changes in global and chromatin-associated OGT were observed between MCF-7 and ADR cells (Fig. 2a, e), suggesting that genotoxic stress-induced O-GlcNAc fluctuations in chromatin are independent of OGT expression.

We subsequently performed COGC-seq with biological repeatability tests to investigate the genome-wide O-GlcNAc loci (Supplementary Fig. 8 and Data 7). The COGC-seq peaks showed distinct characteristics across the genome in MCF-7 and ADR cells (Fig. 2f). Consisting of the identification of more OCPs in ADR cells, the COGC-seq peak height obtained for ADR cells was higher than that found for MCF-7 cells (Fig. 2g). In addition, both MCF-7 and ADR signals were widely distributed at transcription start sites (TSSs) with a sharp single peak. Similar results were obtained by traditional ChIP-seq using the O-GlcNAc-recognized lectin succinyl wheat germ agglutinin (sWGA, Supplementary Fig. 9 and Data 8). The data obtained using the chemical reporter-based O-GlcNAc chromatin enrichment strategy indicated that genotoxicity-induced O-GlcNAc enhances the interaction of OCPs with chromatin.

**OCTFs undergo an enhancer-promoter binding switch and dynamically activate diverse target genes in the response to genotoxic stress**. We then investigated how O-GlcNAc affects target gene expression and observed a shift in the COGC-seq signal from a promoter-and-intron/intergenic-balanced distribution (MCF-7) to a promoter-biased distribution (ADR) during the genotoxic stress response (Fig. 3a, b and Supplementary Data 7). These data indicate that genotoxic stress might cause the loss of multiple OCTFs at enhancer elements and induce a genome-wide preferential association of OCTFs at specific gene TSSs. To further elucidate how O-GlcNAz sites (COGC-seq peaks) change during the course of the genotoxic stress response, we examined the dynamics of O-GlcNAz sites using MAnorm[33], which facilitates quantitative comparisons of peaks derived from two pairwise datasets (Supplementary Fig. 10a–c, Data 9). Although a large fraction of peaks persisted between MCF-7 and ADR cells (we refer to such peaks as "unbiased"), we identified 3367 differential quantitative peaks that were either lost or gained during the response to genotoxic stress ("MCF-7-biased" or "ADR-biased", Fig. 3c), compared with the baseline scenario. Similar findings were obtained using another differential binding analysis tool, DiffBind[34] (Supplementary Fig. 10d–g and Data 9).

Previous studies have suggested that promoters and enhancers typically flanked by histone H3 trimethylated at lysine 4 (H3K4me3) and histone H3 monomethylated at lysine 4 (H3K4me1), respectively, and both are additionally marked by histone H3 acetylated at lysine 27 (H3K27ac) upon activation[35]. To confirm the switch in the COGC-seq distribution between enhancers to promoters, we measured these histone markers at genotoxic stress-biased COGC-seq peaks using published ChIP-seq datasets generated from MCF-7 cells. Overall, the COGC-seq datasets shared a high degree of conservation with transcriptionally activate chromatin and O-GlcNAc genomic loci in other cells

(Supplementary Fig. 11a). A low signal for histone H3 trimethylated at lysine 27 (H3K27me3, repressive mark)[36] was found throughout the O-GlcNAz sites in both MCF-7 and ADR cells, whereas a high H3K27ac signal[37] was measured in all these regions, which suggests that the binding of OCTFs is associated with transcriptional activation (Fig. 3d). MCF-7-biased regions were surrounded by the highest levels of H3K4me1[36] and lowest levels of H3K4me3[37], which indicates that the unique OCTFs of MCF-7 cells play a predominant regulatory role at enhancers rather than promoters. In contrast, unbiased and ADR-biased sites showed distinct increases in H3K4me3 levels, whereas the H3K4me1 levels were reduced at these sites. These patterns were also observed in analyses of promoter and intron/intergenic regions (Supplementary Fig. 11b). Genotoxicity reduced the O-GlcNAz sites associated with reported enhancer and super-enhancer elements in MCF-7 cells[38] (Fig. 3e, f). The O-GlcNAz signals at these elements in ADR cells were lower than those in MCF-7 cells (Fig. 3g, h). These data suggest that OCTFs undergo an enhancer-promoter binding switch and are associated with transcriptional activation under genotoxic stress conditions.

Because the COGC-seq signal distributions showed the typical TF-bound characteristic (Fig. 2g), we subsequently screened candidate OCTFs. By scanning MCF-7- and ADR-biased COGC-seq peaks using motif discovery algorithms (Homer[39]), we found that diversified TF-binding sites were enriched in MCF-7 and ADR-biased regions (Fig. 4a). After accounting for the putative TF-binding sites and targeting genes in all O-GlcNAz sites, the majority of the peaks contained numerous TF-binding sites and exhibited distinct patterns between MCF-7 and ADR-biased regions (Fig. 4b and Supplementary Data 10). An appreciable number of 367 putative TFs were identified in the differential quantitative COGC-seq peaks (Supplementary Data 11). Among them, 33 candidate TFs overlapped with 88 differential OCTFs identified from the proteomics analysis (Fig. 3c, Supplementary Data 12). Certain OCTFs were validated by ChIP-qPCR (Supplementary Fig. 12). Correspondingly, the overlapping OCTF-targeting genes (1550) were categorized based on GO terms related to the stress response, DNA damage and DNA repair. Similar results were obtained for the OCTF-targeting genes in MCF-7 and ADR cells (Supplementary Fig. 13 and Data 13).

Based on the analysis of the differential COGC-seq peaks between MCF-7 and ADR cells, 1572 O-GlcNAz-associated genes (including a number of unreported O-GlcNAc-targeting genes) were uniquely found in MCF-7 cells, whereas 1042 genes exclusive to ADR cells were observed. Notably, only 27 genes were annotated in both MCF-7- and ADR-biased peaks (Fig. 4d and Supplementary Data 7). A GO analysis revealed that the genes associated with ADR-biased peaks were more enriched in roles associated with the stress response than those associated with MCF-7-biased peaks (Fig. 4e). Representative results from the visualization and verification of discrete genomic loci occupied by O-GlcNAz, including the stress-related genes *DNAJA1*[40], *PPP2R5B*[41], *PDCL3*[42], and *MAN2C1*[43], illustrate the COGC-seq peak changes at the individual gene level (Fig. 4f and Supplementary Fig. 14). Collectively, the genome-wide study of OCPs, particularly OCTFs, revealed an enhancer-promoter binding switch and dynamic activation of stress response-related genes during exposure to genotoxic stimuli.

**An integrative analysis reveals the potential genotoxic stress-responsive transcriptional reprogramming regulated by OCTFs**. To further illustrate the OCTF-regulated gene network, an integrative omics analysis of MCF-7 and ADR datasets was performed (Fig. 5a). The comparison of the RNA-seq DEGs (7112 genes) with differential COGC-seq peak-associated genes (2194 genes) yielded 976 overlapping genes (Supplementary

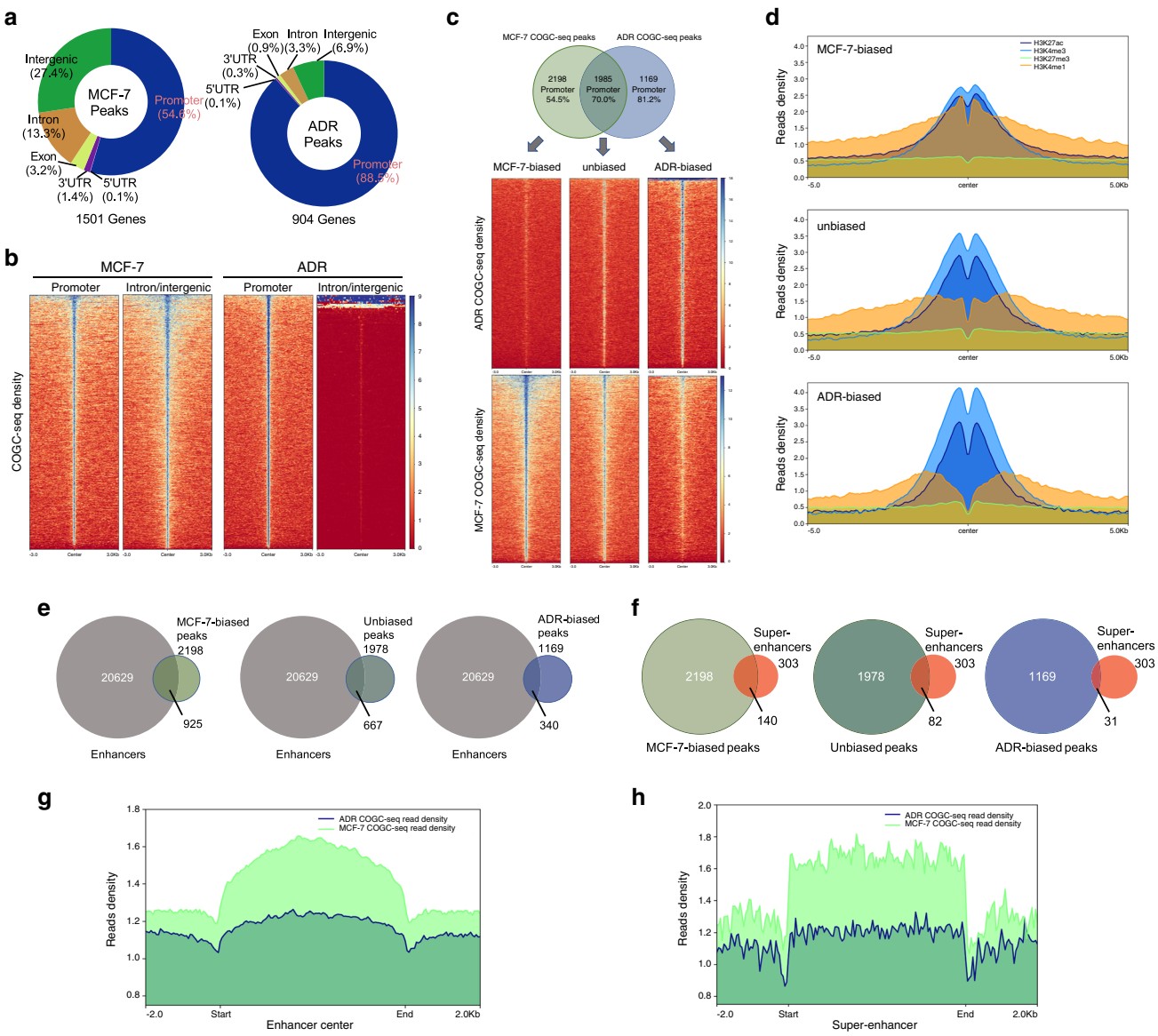

**Fig. 3 OCTFs reveal an enhancer-promoter binding switch during exposure to genotoxic stimuli. a** MCF-7 and ADR COGC-seq peak annotation relative to known genomic elements. **b** Heat map representation of the COGC-seq signal at the promoter and intron/intergenic regions bound by OCPs in MCF-7 and ADR cells. Enrichment levels (red, low; blue, high) were profiled ±3 kb from the peak center. **c** COGC-seq occupancy signal in MCF-7-biased, ADR-biased and unbiased peaks identified by MAnorm (fold change ≥ 2 and $P$ value ≤ $10^{-5}$). Upper panel: Venn diagram showing the overlap of COGC-seq peaks in MCF-7 and ADR cells. The percentage of peaks annotated to promoter regions is indicated. Lower panel: Heat map representation of COGC-seq signal enrichment (red, low; blue, high) at differential quantitative peaks. The enrichment levels were profiled ±3 kb from the peak center. **d** Average enrichment profiles of published H3K27ac, H3K4me3 (GSE97481), H3K27me3 (GSE96363) and H3K4me1 (GSE86714) ChIP-seq reads at differential quantitative COGC-seq peaks. **e, f** Overlap of differential quantitative COGC-seq peaks with MCF-7 enhancer and super-enhancer regions reported previously. **g, h** Average enrichment profiles of COGC-seq reads at MCF-7 enhancer and super-enhancer regions.

Fig. 15). We subsequently linked these genes to the above-mentioned OCTF-targeting genes (1550 genes) and found that 647 genes were directly regulated by 33 diverse OCTFs during adaptation to genotoxic stress (Supplementary Data 14). We established an O-GlcNAc-regulated stress response gene expression network. (Fig. 5b). These data suggest that the top candidate OCTFs, NRF1, SP1 and KLF5 are likely to play a central role. Of note, the majority of genes in the network were enriched in the cellular stress response, apoptosis, and DNA repair, and several of them were unreported terms for the genotoxic response (Fig. 5c, d). Furthermore, the majority of sufficiently expressed genes were occupied by OCTFs in both MCF-7 and ADR cells

(Supplementary Figure 16), suggesting that the genomic binding of OCTFs activated gene transcription (Fig. 3d).

To demonstrate the role of O-GlcNAc in stress-induced gene transcription and the cellular phenotype associated with adaptation to genotoxic stress, we conducted mRNA and cell viability analyses in the presence of the O-GlcNAc inhibitor L01[44] or agonist PugNAc[45]. The transcription of the OCTF-targeting genes *THUMPD3*, *OTUD7B*, *MAN2C1*, *SEC13*, and *PPP2R5B* was attenuated in ADR cells treated with L01, whereas the expression of these genes were accumulated in PugNAc-stimulated MCF-7 cells (Fig. 5e). The loss of the important OCTFs SP1 and KLF5 also decreased downstream gene transcription and ADR cell

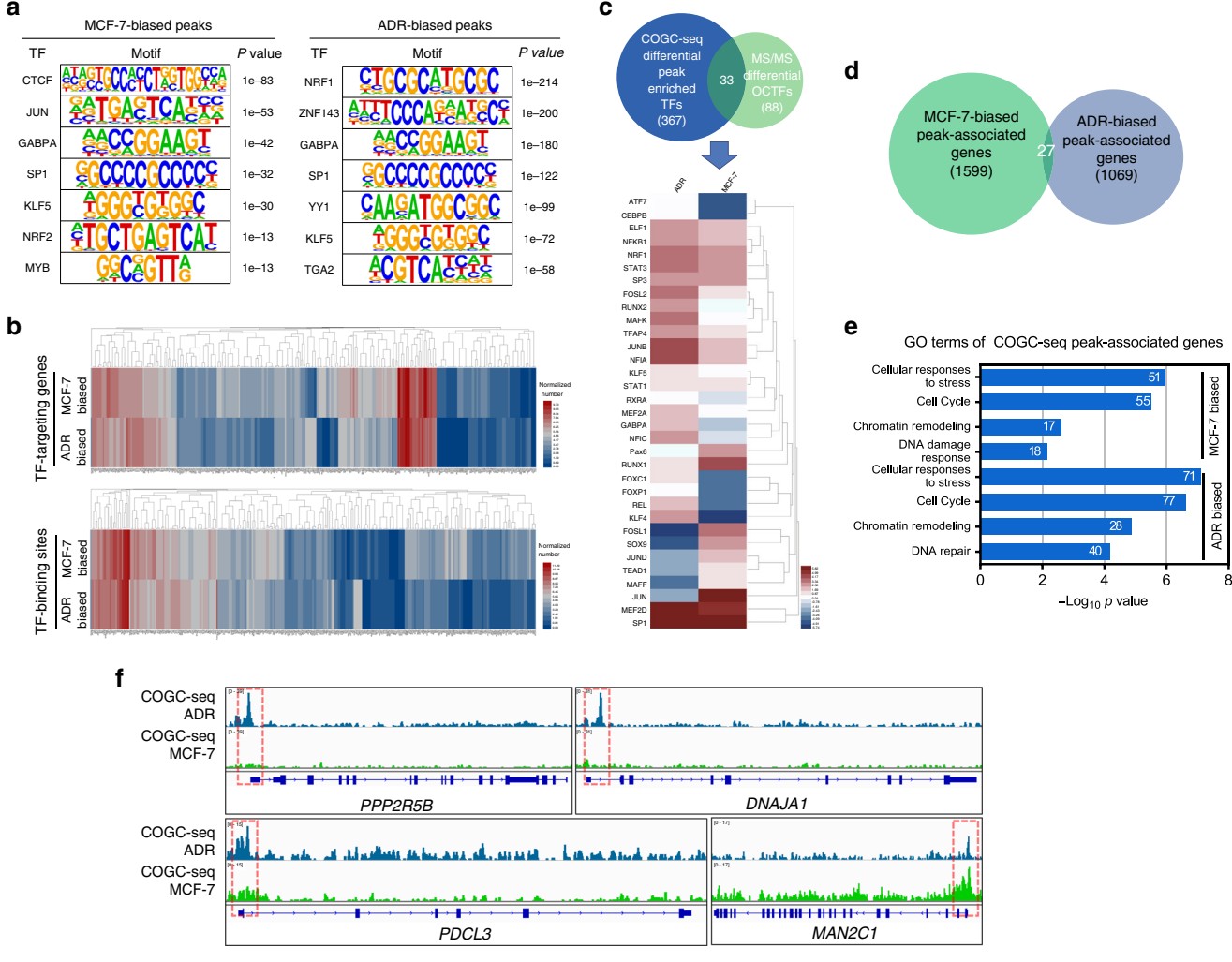

**Fig. 4 OCTFs dynamically target stress response genes during the genotoxic response. a** Motif discovery analysis of differential quantitative COGC-seq peaks using Homer. The binomial test *P* value of each motif is shown. **b** The chromatin binding sites and target genes of 367 putative TFs generated by the motif discovery analysis of COGC-seq peaks are shown in heat maps (red, high; blue, low). **c** Overlap of 367 enriched TFs identified from the motif discovery analysis with 88 differential quantitative OCTFs in MS/MS is shown in a Venn diagram. The quantification of the 33 overlapping O-GlcNAc TFs is shown in a heat map (red, high; blue, low). **d** Venn diagram showing the overlapped peak-associated genes identified from MCF-7 and ADR-biased COGC-seq regions. **e** GO analysis of MCF-7 and ADR-biased peak-associated genes. The gene numbers identified in certain terms are shown. Source Data are provided as a Source Data file. **f** Integrative Genomics Viewer (IGV) tracks showing COGC-seq signal at the promoter regions of the representative genes *PPP2R5B*, *DNAJA1*, *PDCL3*, and *MAN2C1*.

viability, which suggest that these genes are directly regulated by O-GlcNAc SP1 and KLF5 during the response to genotoxic stress (Supplementary Fig. 17). Moreover, L01 significantly decreased the viability of ADR cells upon genotoxic provocation, whereas PugNAc induced a cytotoxic reduction in Adm-stimulated MCF-7 cells (Fig. 5f). Taken together, the present data expand our knowledge of the full range and spectrum of genotoxic stress-related gene transcription that directly requires OCTFs.

**O-GlcNAc modification of the response regulator NRF1 contributes to intensive chromatin occupation during exposure to genotoxic stress.** Because OCTFs play key regulatory roles during the response to genotoxic stress, we subsequently explored the central factors that contribute to the alternation of so many gene expression. OCTFs were ranked based on their motif enrichment values and their number of genome-wide binding sites (Fig. 6a). Although the three most enriched central OCTFs were identified,

little difference in the enrichment, binding sites and target genes of SP1 and KLF5 was observed between MCF-7- and ADR-biased COGC-seq peaks. SP1 and KLF5 belong to the same TF family and bind to similar DNA motifs[45]. The representative SP1 ChIP-seq signal[36] showed similar levels across differential COGC-seq peaks (Supplementary Fig. 18a), which indicates that SP1 and KLF5 cannot be the key factors during the response to genotoxic stress. In contrast, substantial changes in the chromatin-binding parameter and ChIP-seq signal distribution in MCF-7- and ADR-biased peaks were detected for nuclear respiratory factor 1 (NRF1, Uniprot ID: Q16656) (Fig. 6a, b and Supplementary Fig. 18b). Consistent with this finding, 9.4% of MCF-7 COGC-seq peaks were occupied by NRF1, whereas 45.6% of the ADR peaks overlapped with NRF1 peaks (Fig. 6c), which suggests the influential role of this TF in the O-GlcNAc-regulated genotoxic stress response.

We then examined the O-GlcNAc state of NRF1 during Adm-induced cellular stress. O-GlcNAc on NRF1 could be detected

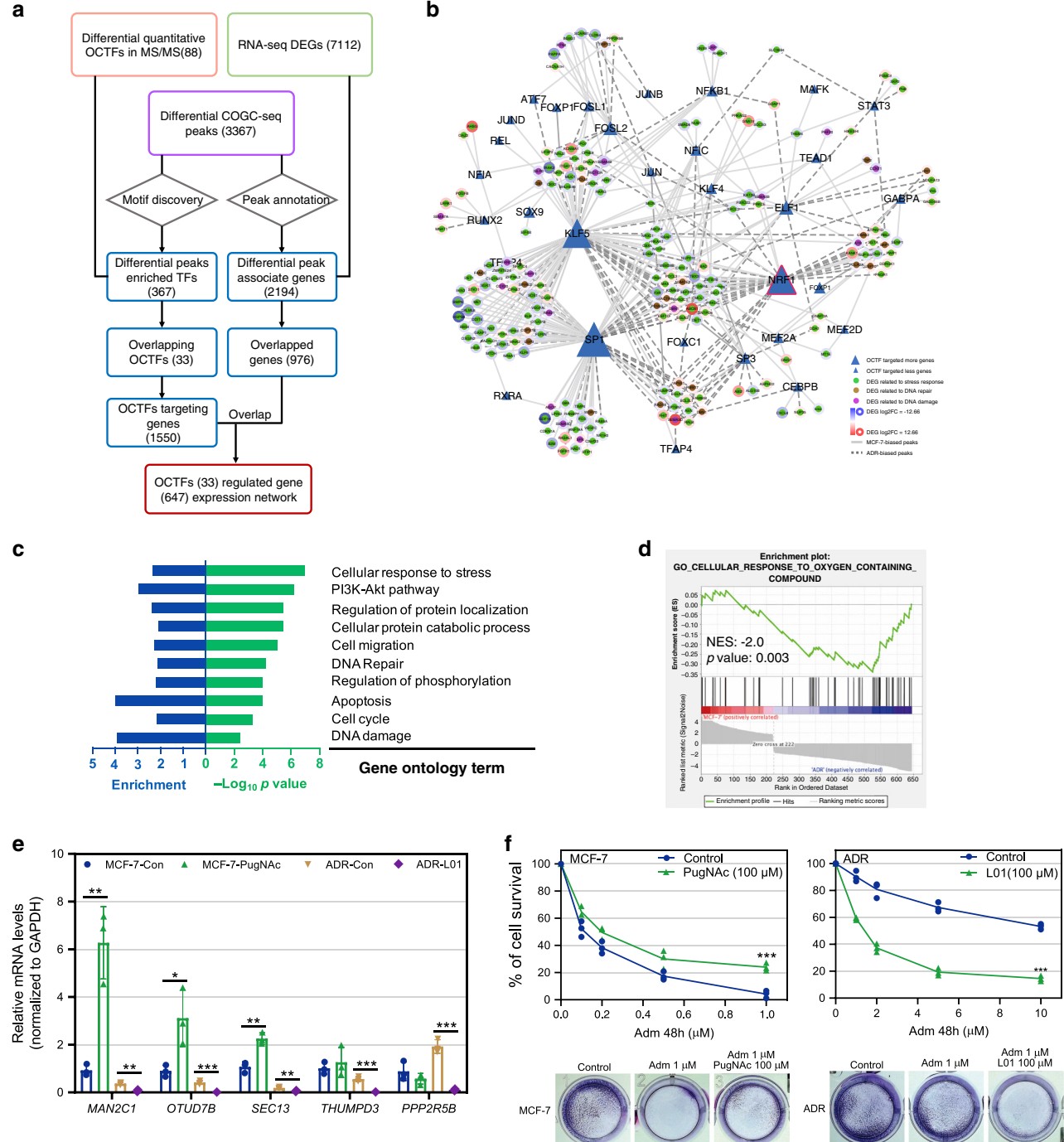

**Fig. 5 OCTFs regulate genotoxic stress-responsive transcriptional reprogramming. a** Schematic of an integrative omics analysis strategy for the discovery of an OCTF-regulated gene expression network during the genotoxic response. RNA-seq DEGs of MCF-7 and ADR cells were compared with COGC-seq differential peak-associated genes. The overlapping genes were subsequently linked to OCTF-targeting genes. The end result was 33 genotoxic stress-response OCTFs that directly regulate the expression of 647 genes to maintain cellular homeostasis. **b** Predicted regulatory network between genotoxic stress-responsive OCTFs (triangle nodes) and the corresponding downstream DEGs (round nodes) in MCF-7 and ADR cells. The solid and dashed lines represent regulatory relationships from OCTFs to target genes in MCF-7 and ADR cells, respectively. Highly connected OCTFs are indicated with a large size. For the round nodes, different round fill colors represent the supposed function of the DEGs and the different border colors represent the expression level of DEGs (red, high; blue, low). **c** GO terms enriched for 647 OCTFs targeting genes in the network. **d** GSEA of OCTF-targeting genes in the network revealing enrichment in pathways related to the stress response. The NES and *p* value are indicated. **e** Effect of 100 μM PugNAc or L01 (24 h) on the indicated gene mRNA levels in MCF-7 and ADR cells. Con, control (vehicle). The data are presented as means ± SEM. **f** MCF-7 and ADR cells were treated with increasing doses of Adm alone (control, Con) or together with 100 μM L01 (or PugNAc) for 48 h, and the cell viability was assessed using CCK8 assay. Replicates are represented. Representative images of cell viability determined by crystal violet staining are shown. The staining results were reproduced in two biologically independent experiments. For (**e**) and (**f**), *n* = 3 biologically independent experiments. *$p < 0.05$, **$p < 0.01$, ***$p < 0.001$ (two-sided unpaired Student's *t*-test). *p* values: 0.003794, 0.001189, 0.033775, 0.000744, 0.003298, 0.005793, 0.000656, 0.00039 (**e**); 0.000991, 0.000021 (**f**). Experiments were repeated independently two times (**e**) and (**f**) with similar results. **c**, **e**–**f** Source Data are provided as a Source Data file.

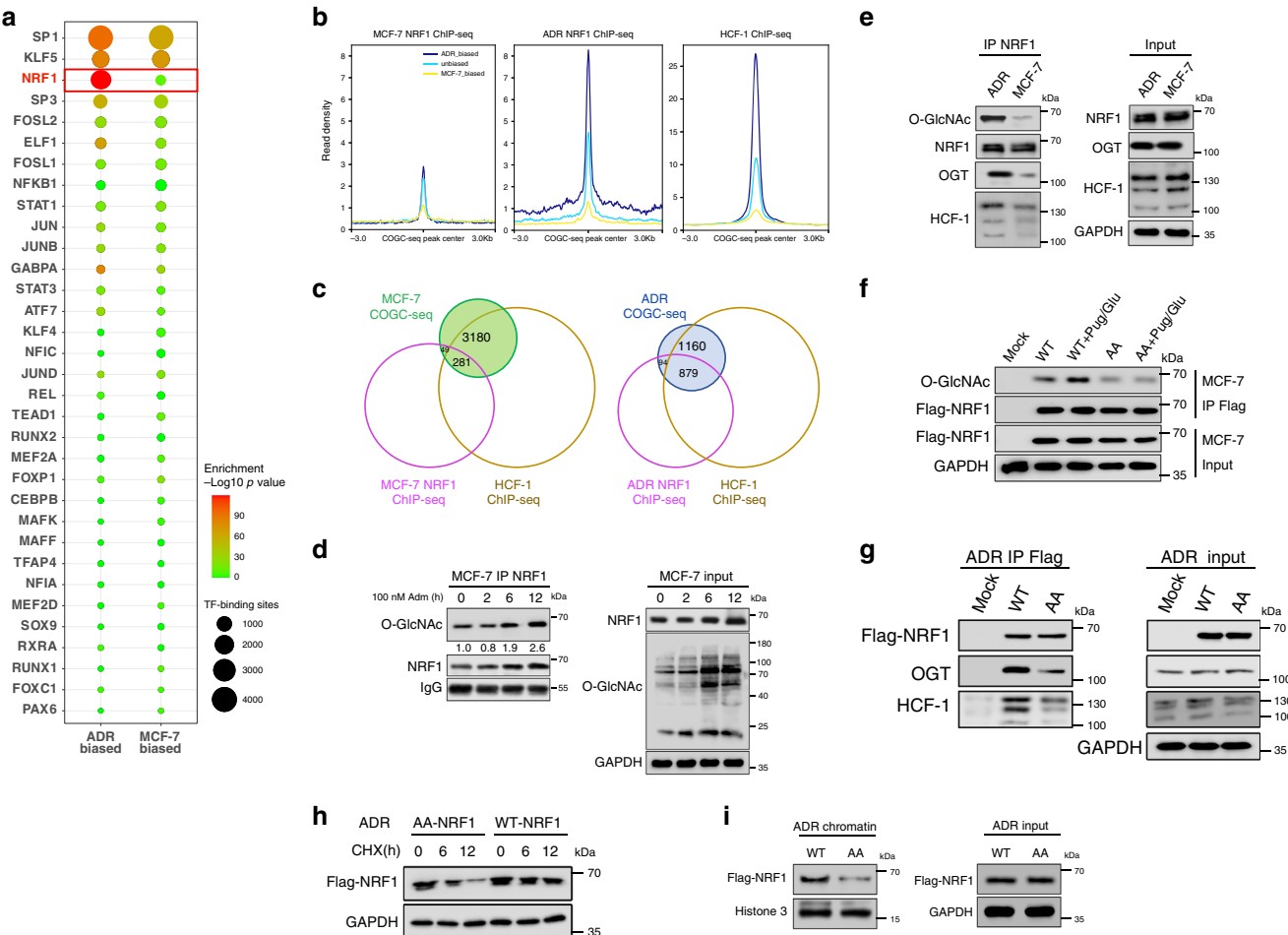

**Fig. 6 O-GlcNAc enhances the stability and chromatin binding of NRF1 during the response to genotoxic stress. a** The chromatin binding sites and enrichment of 33 responsive OCTFs generated through a motif enrichment analysis of MCF-7- and ADR-biased COGC-seq peaks are shown in a heat map. The color of the dots represents the TF motif enrichment. More TF-binding sites are indicated by a large dot size. **b** Average enrichment profiles of NRF1 ChIP-seq reads (MCF-7 and ADR cells) and published HCF-1 ChIP-seq reads (MCF-7, GSE91992) at differential quantitative COGC-seq peaks. **c** COGC-seq peaks in MCF-7 and ADR cells overlap with NRF1 and a published HCF-1 (GSE91992) ChIP-seq dataset. **d** O-GlcNAc NRF1 was upregulated in MCF-7 cells after transient stimulation with 100 nM Adm. IP of NRF1 was performed, and the immunoprecipitated fractions were analyzed by immunoblotting for O-GlcNAc (CTD110.6). **e** The NRF1-HCF-1 interaction is increased in ADR cells compared with MCF-7 cells. NRF1 co-IP was performed, and the immunoprecipitated fractions were analyzed by immunoblotting for the indicated proteins. **f** NRF-1 is O-GlcNAcylated at Ser448/Ser451. After treatment with PugNAc (Pug, 100 µM) and glucose (Glu, 25 mM) for 24 h, MCF-7 cells stably expressing Flag-WT-NRF1 or Flag-AA-NRF1 were immunoprecipitated with anti-Flag magnetic beads. O-GlcNAcylation (CTD110.6) was analyzed by immunoblotting. WT, wild-type NRF1; AA, Ser447/Ser450 → Ala mutational NRF1. Mock, cells transfected with empty pCMVPuro64 vector. **g** O-GlcNAc promotes the interaction of HCF-1 and OGT with NRF1. Immunoblotting showing the PPIs of endogenous HCF-1 and OGT with NRF1 in ADR cells. ADR cells stably expressing Flag-WT-NRF1 or Flag-AA-NRF1 were immunoprecipitated with anti-Flag magnetic beads. **h** O-GlcNAc inhibition expedites the degradation of NRF1. ADR cells expressing Flag-WT-NRF1 or Flag-AA-NRF1 were incubated with 50 µM cycloheximide (CHX) for up to 12 h. The expression levels of Flag-NRF1 were monitored by immunoblotting. **i** O-GlcNAc enhances the chromatin binding of NRF1. The crosslinked chromatin proteins were extracted, and the levels of Flag-NRF1 were detected by immunoblotting. For (**d**–**i**), all blots are representative of at least two biologically independent experiments. **a**, **d**–**i** Source Data are provided as a Source Data file.

with an O-GlcNAc-specific antibody after 6 h of Adm stimulation, and this detection was accompanied by the accumulation of total NRF1 (Fig. 6d). The amount of modified NRF1 in MCF-7 cells treated with Adm was significantly higher after 12 h, which indicates the potential regulatory function of O-GlcNAc at this time point. To further eliminate the nonspecific immunostaining of the anti-O-GlcNAc antibody, an O-GlcNAc enzymatic labeling system[46], which recognizes terminal GlcNAcs with high specificity, was employed to confirm the modification of O-GlcNAc on endogenous NRF1 in MCF-7 and ADR cells (Supplementary Fig. 19).

IP of these cells showed that O-GlcNAc increased the PPIs between NRF1 and HCF-1, which is a previously reported OGT-

binding partner[47] (Fig. 6e). The interaction between NRF1 and OGT revealed the same changing tendency. These observations were also confirmed by the changes in the HCF-1 ChIP-seq signal at the O-GlcNAz sites (Fig. 6b, c). To disclose the potential O-GlcNAc modification sites of NRF1, we stably introduced 3× Flag-NRF1 predicted mutated at two predicted O-GlcNAc sites (YinOYang 1.2 Server, cbs.dtu.dk/services/YinOYang) into MCF-7 and ADR cells (AA-NRF1 with Thr342/Thr500→Ala according to the human NRF1 sequence). The O-GlcNAc level of chromatin-bound AA-NRF1 was significantly reduced compared with that of wild-type NRF1 (WT-NRF1) in the absence or presence of PugNAc and glucose, which suggests that the majority of chromatin-bound NRF1 O-GlcNAc modification occur on Thr342/Thr500 in this cell

model (Fig. 6f). Our test also showed that low O-GlcNAc levels reduced the interaction of NRF1 with HCF-1 and OGT (Fig. 6g). Moreover, O-GlcNAc stabilized NRF1 by attenuating its ubiquitin-dependent degradation (Fig. 6h and Supplementary Figure 20). Notably, even though the two types of recombinant NRF1 had the similar expression levels, AA-NRF1 showed significantly less chromatin binding than WT-NRF1 (Fig. 6i), which indicates that the abundance of chromatin-bound NRF1 is monitored by O-GlcNAc modification. Furthermore, the stable knockdown of NRF1 significantly increased the genotoxicity of Adm to ADR cells. Cell death was reversed by the rescue expression of WT-NRF1 but not AA-NRF1, which indicates the key role of O-GlcNAc NRF1 in adaptation to genotoxic stress (Supplementary Fig. 21a, b). Therefore, O-GlcNAc could enhance the stability of NRF1 and promote its dynamic assembly with chromatin during the genotoxic stress response.

**O-GlcNAc NRF1 is essential for the transcriptional activation of various genes to maintain protection against genotoxic stimuli.** The comparison of the NRF1 ChIP-seq peaks in ADR and MCF-7 cells revealed that more than 5000 unique binding sites, in addition to those already occupied in MCF-7 cells, showed increased NRF1 binding after genotoxic adaptation (Fig. 7a, Supplementary Fig. 22 and Data 15). Uniquely bound NRF1 sites correlated with ADR COGC-seq signal, and a high H3K27ac signal[37] was also detected in these regions (Fig. 7b, c). A large fraction of these sites (55.4%) occurred at promoter regions in ADR cells, which suggests that genotoxicity-induced O-GlcNAc enhances NRF1 transcriptional regulatory function.

Consistently, the increase in NRF1 binding was also found to match by a significant increase in the expression of downstream genes (Figs. 6b and 7d). The elimination of NRF-1 O-GlcNAc (AA-NRF-1) significantly attenuated chromatin binding compared with the elimination of WT-NRF1, and this finding was observed in both stably transfected MCF-7 and ADR cells (Fig. 7e). The expression levels of the representative NRF1 unreported targets *NSMCE2*[48],

*DNAJA1*[40], *JAG2*[49], and *PDCL3*[42], which are important for stress tolerance, in cells expressing AA-NRF1 were markedly downregulated compared with those in WT-NRF1-expressing ADR cells (Fig. 8a). Accordingly, O-GlcNAc inhibition and down-regulation of NRF1 also suppressed the expression of these genes and the protection against Adm stress in ADR cells (Supplementary Fig. 21). Furthermore, both the ChIP-seq and ChIP-qPCR results revealed that the promoter regions of these representative genes were robustly bound by WT-NRF1 but weakly bound by AA-NRF1 (Fig. 8b), which suggests that a pool of candidate genes could be upregulated by O-GlcNAc NRF1.

To corroborate these findings, we turned our attention to an unreported O-GlcNAc NRF1-promoted gene, *NSMCE2*, which was recently found to be associated with DNA repair[48]. The luciferase activity of the *NSMCE2* promoter was significantly reduced after treatment with the O-GlcNAc inhibitor L01 in WT-NRF1 expressing 293T cells. Accordingly, AA-NRF1 exhibited minimal transcriptional activity, which supports that the O-GlcNAc modification of NRF1 enhances its transcriptional activity (Fig. 8c). Using NRF1 shRNA, we confirmed that the stress-induced increase in the mRNA and protein levels of *NSMCE2* was dependent on NRF1 (Supplementary Fig. 21c). Furthermore, *NSMCE2* knockdown visibly reduced the protection of ADR cells from Adm, which indicates the adaptive role of this gene in genotoxicity (Fig. 8d). An overall survival analysis showed that a high expression of *NSMCE2* was associated with a worse outcome in breast cancer patients (Supplementary Fig. 23).

Taken together, these data demonstrate that the activation of NRF1 by O-GlcNAc modification could elevate the transcription of various genes to protect against genotoxic stimuli. O-GlcNAc is likely to function as a stress sensor that strengthens the genotoxic resistance of cancer cells.

## Discussion

Different from the classical types of glycosylation, hundreds of nucleocytoplasmic proteins undergo O-GlcNAc modification, in

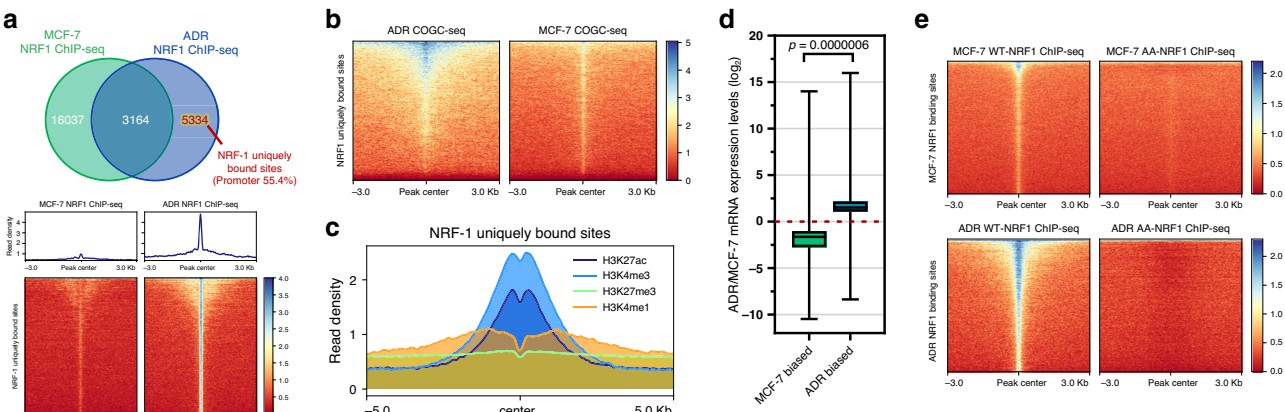

**Fig. 7 Genotoxicity-induced O-GlcNAc enhances NRF1 transcriptional regulatory function. a** NRF-1 ChIP-seq signal in NRF-1 uniquely bound sites identified by overlapping MCF-7 and ADR NRF-1 peaks. Upper panel: Venn diagram showing the overlap of NRF-1 peaks in MCF-7 and ADR cells. The percentage of peaks annotated to promoter regions is indicated. Lower panel: Heat map representation of NRF-1 signal enrichment (red, low; blue, high) at NRF-1 uniquely bound sites. The enrichment levels were profiled ±3 kb from the peak center. **b** Heat map representation of COGC-seq signal enrichment (red, low; blue, high) at NRF-1 uniquely bound sites. The enrichment levels were profiled ±3 kb from the peak center. **c** Average enrichment profiles of published H3K27ac, H3K4me3 (GSE97481), H3K27me3 (GSE96363) and H3K4me1 (GSE86714) ChIP-seq reads at NRF-1 uniquely bound sites. **d** The box plots showing the mRNA expression changes (RNA-seq FPKM) of NRF1-binding genes associated with MCF-7- and ADR-biased peaks. The box plots show the medians (black lines), 25th and 75th percentiles (boundaries), and minimum/maximum values (whiskers). The *p* value (0.0000006, two-sided unpaired Student's *t*-test, calculated between multiple genes in each group) is indicated. *n* = 2 biologically independent RNA-seq replicates. Source Data are provided as a Source Data file. **e** Heat map representation of WT-NRF-1 and AA-NRF-1 signal enrichment (red, low; blue, high) at NRF-1 binding sites in MCF-7 and ADR cells. The enrichment levels were profiled ±3 kb from the peak center.

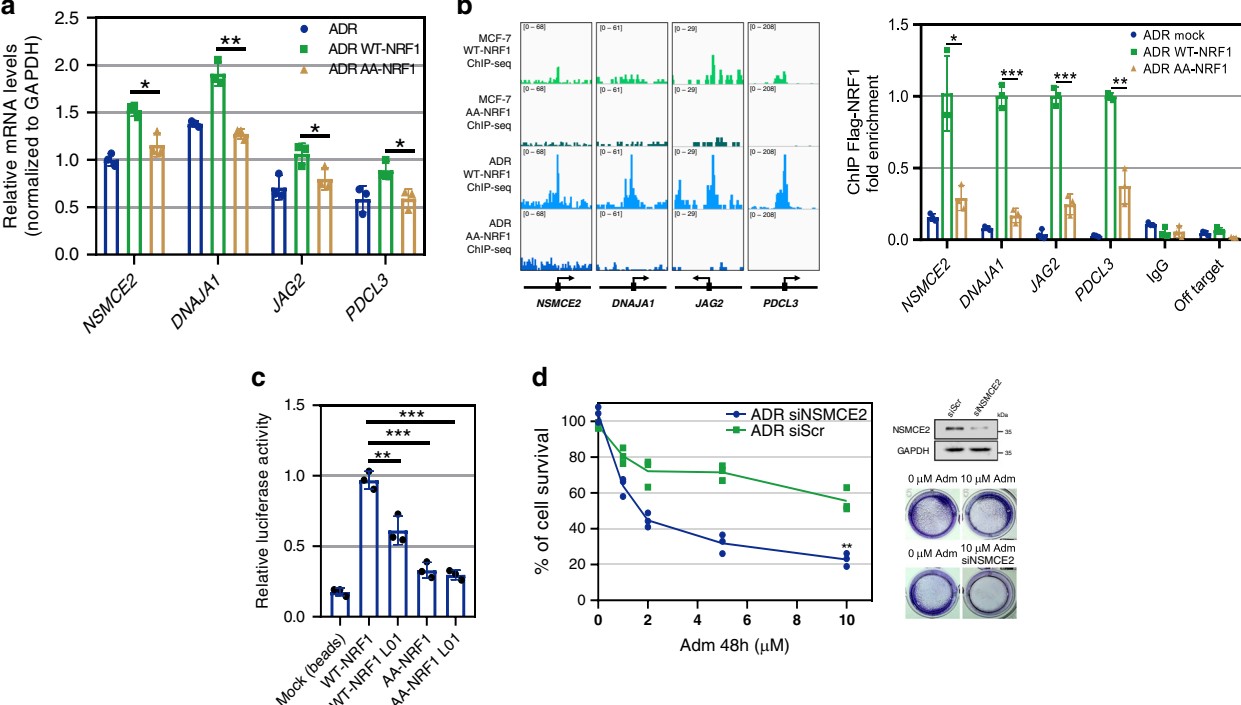

**Fig. 8 O-GlcNAc NRF1 increases the pool of gene transcription to protect cells against genotoxicity. a** Effect of NRF1 O-GlcNAc modification on the indicated gene transcription levels in ADR cells. The gene mRNA levels in ADR cells expressing Flag-WT-NRF1 or Flag-AA-NRF1 were analyzed by quantitative PCR (qPCR). **b** Left panel: IGV tracks showing the signals at the promoter regions of the representative genes. Right panel: Validation of O-GlcNAc NRF1 binding peaks by ChIP-qPCR. qPCR amplification was performed. Each bar represents the fold enrichment of binding relative to the input. IgG and random primers that could not specifically bind the indicated gene promoter regions (off target) were used as negative controls. Mock, cells transfected with empty pCMVPuro64 vector. **c** O-GlcNAc inhibition reduces the transcriptional activity of NRF1. 293T cells were transfected with a reporter vector consisting of luciferase cDNA fused to the *NSMCE2* promoter. The pGL3-basic vector (Mock) was used as a control. **d** ADR cells were transfected with *NSMCE2* siRNA (siNSMCE2) or scrambled siRNA (siScr) and treated with increasing doses of Adm for 48 h. The cell viability was then assessed. Representative images of cell viability determined by crystal violet staining are shown. Results were reproduced in two biologically independent experiments. The protein levels of NSMCE2 were monitored by immunoblotting. All blots are representative of at least two biologically independent experiments. For (**a**–**c**), the data are presented as the means ± SEM., (**d**) replicates are represented. (**a**–**d**) $n = 3$ biologically independent experiments, *$p < 0.05$, **$p < 0.01$, ***$p < 0.001$ (two-sided unpaired Student's *t*-test). *p* values: 0.010099, 0.001375, 0.046814, 0.020148 (**a**); 0.010191, 0.000117, 0.000159, 0.001032 (**b**); 0.0065 (WT-NRF1 vs. WT-NRF1 L01), 0.0002 (WT-NRF1 vs. AA-NRF1), 0.0000858 (WT-NRF1 vs. AA-NRF1 L01) (**c**); 0.001575 (**d**). **a**–**d** Experiments were repeated independently two times with similar results. Source Data are provided as a Source Data file.

which the monosaccharide N-acetylglucosamine moiety is neither epimerized nor elongated. As a reversible posttranslational modification, O-GlcNAc dynamically orchestrates the activities and functions of a wide range of proteins in multiple pathways to maintain cellular homeostasis[1]. Other research groups and ours have demonstrated that an acute elevation of the global O-GlcNAc modification triggers an endogenous protective procedure in various models of diverse environmental cues, including genotoxicity[11]. However, the mechanisms that accurately represent the reciprocal interplay between O-GlcNAc and stress pathways remain unclear.

Transcriptional reprogramming is one of the crucial steps that occur during stress responses[19]. Interestingly, increasing lines of evidence confirm that stress-responsive transcription regulators are modified by O-GlcNAc[17,18], which indicates the participation of this glycosylation in gene expression during stress adaptation. Because distinct groups of TFs and target genes drive a common response process, such O-GlcNAc-mediated regulatory modes should exhibit functionally coordinated associations, including O-GlcNAc TFs and relationships between TF genomic binding loci and gene transcription. However, individual experiments that simply analyze subsets of regulators cannot expose the full O-

GlcNAc regulatory landscape of cellular homeostasis. Integrative surveying of multilayered data would help uncover the O-GlcNAc-regulated stress response gene expression network.

In the present study, a clickable unnatural monosaccharide GalNAz was employed for the metabolic labeling of cellular O-GlcNAc proteins in living cells and subsequent chemoselective conjugation with a bioorthogonal biotin group for O-GlcNAz chromatin-specific isolation and hierarchical analysis. The use of this efficient and specific O-GlcNAc labeling reporter was recently certified by the absence of or minimal artificial S-glycosylation appearance[27,50]. Because GalNAz might also label cell-surface N-linked and mucin-type O-linked glycans, we isolated nuclei and separated crosslinked chromatin from metabolically labeled cells. These enriched O-GlcNAz chromatin could provide materials for occurrent OCP profiling and interrelated genome-wide DNA sequencing. Using this strategy, along with the dynamic transcriptome, multilayered omics datasets could be obtained and could be powerful for elucidating genotoxic stress-induced O-GlcNAc-regulated gene expression network with broad coverage and context-specific dynamics.

Because the response to Adm provides a versatile model for investigating genotoxic stress-influenced transcriptional

regulation[31,32], MCF-7 cells and the genotoxic-adaptive variant ADR cells were labeled with GalNAz. To eliminate the possibility that the LC-MS/MS samples were contaminated with nonlabeled proteins due to formaldehyde-induced PPI crosslinking, the O-GlcNAz chromatin was effectively decrosslinked as previously reported[28] prior to OCP capture. The enriched OCPs were subjected to a strict washing process to eliminate nonspecific contamination. After rigorous valid value filtering, our quantitative analysis of the OCP subproteome revealed alterations in multiple transcription regulators, which cover various functional categories, and these regulators exhibited high enrichment in the stress response and DNA repair. Here, only the differential quantitative OCPs located in the nucleus were identified as regulatory candidates, which further reduced the effect of nonspecific contamination during the OCP capture step on the ultimate network construction and analysis. The specificity of this chemical reporter-based OCP enrichment was also verified by the distinct difference between the OCP quantity and gene expression in RNA-seq and whole-cell proteomics. Recent reports have suggested that chromatin remodeling complexes are undergoing O-GlcNAc modification[51–53]. Not surprisingly, numerous members of the chromatin remodeling complex, including SIN3A, NuRD, and TRAP/SMCC, were identified in the OCP subproteome, which indicated that O-GlcNAc has a function in modulating chromatin conformation. Moreover, consistent with the elevation of cellular O-GlcNAc in response to various stressors[11,13], our data showed that most OCTFs (such as the well-known O-GlcNAc TFs SP1[54], NFκB[55], and SIN3A[51]) increased their interactions with chromatin rather than decreased chromatin binding in Adm-adaptive cells, which suggested the strong role of O-GlcNAc in the binding of OCPs to the corresponding genomic loci. Therefore, we propose that OCPs act as stress sensors that have been implicated in genotoxic adaptation.

We subsequently surveyed the dynamics of genome-wide O-GlcNAz sites by COGC-seq. Overall, the genomic loci and intensity of the COGC-seq peaks showed marked variations between the two cell models, which supported the alteration in OCTFs in this study. Although sWGA and O-GlcNAc antibodies have recognized limitations, more O-GlcNAc sites were found using these two methods than by COGC-seq. We speculate that this finding was obtained because diverse OCPs with different turnover rates would be labeled with O-GlcNAz to different degrees[30], and newly biosynthesized proteins could be labeled with higher efficiency. A strong signal correlation was revealed in the comparison of COGC-seq with sWGA lectin ChIP-seq and other O-GlcNAc antibodies ChIP-seq. These results demonstrate the applicability of this metabolic labeling approach in TF-associated DNA enrichment. Previous evidence obtained from *Drosophila* revealed that O-GlcNAc is distributed to specific genomic loci containing polycomb group response elements and thus plays a role in repressing gene expression and regulating epigenetic chromatin modification[24,25]. In contrast, MCF-7 and ADR COGC-seq signals are distributed at transcriptionally active regions across the genome and are clearly distinguished from those of polycomb-repressive marker H3K27me3 sites. These data suggest that the coordination of OCTFs with transcription active chromatin appears to play a principal role in human cells.

Further comparisons between MCF-7 and ADR COGC-seq signals provide insight into the differences in the occupancy of OCTF chromatin elements. Although a "spike-in" analysis should be powerful for identifying global changes in OCP binding, its applicability is limited in COGC-seq. As a prerequisite, only if the foreign reference genome applied in COGC-seq is labeled with azide will the subsequent chemoselective enrichment procedures proceed. Moreover, experimental variations and the discordance

of species-related biorthogonal specificity might also be introduced in COGC-seq, which would affect subsequent data interpretation. Therefore, two statistical ChIP-seq normalization approaches, MAnorm and DiffBind, which were extensively used in recent studies[56–58], were employed to detect genomic regions showing differential quantitative OCP binding. The genotoxicity-induced differential binding regions determined by MAnorm and DiffBind were of a similar magnitude and highly coincident, reflecting the reliability of the results. Using these data, we highlight a genotoxicity-induced enhancer-promoter switch in OCTF genome binding sites. The results presented here support a regulatory mechanism in which O-GlcNAc and TFs coordinate enhancer-promoter switching to reprogram the expression of genes needed for the maintenance of cellular homeostasis. The shift from a balanced promoter/enhancer pattern of O-GlcNAz sites to a promoter-heavy distribution after genotoxic adaptation also provides evidence showing that O-GlcNAc might play a role in chromatin conformation and the occupancy of other regulators. However, the precise mechanisms driving this extensive switch remain unclear and should be considered in future studies.

Combining differential OCTFs and their genomic binding loci in two cell models with DEGs from RNA-seq datasets, we investigated the relevance of multiple OCTFs, concurrent genome binding and target gene transcription levels. The majority of differential COGC-seq peak intensities correlate with the target gene transcript levels. A large fraction of OCTFs identified in our study elevated the chromatin interaction protein quantity as well as binding sites in ADR cells. However, a small decrease in the amount of OCTF binding occurs during exposure to genotoxic stress, which reveals the dynamic state of OCTF genomic binding and the complexity of O-GlcNAc-regulated gene transcription. By integrating multilayer datasets, a network of OCTFs and downstream genotoxic-response genes was established. These results provide a framework of genotoxic stress-induced transcriptional adaptation based on multiple OCTFs in cancer cells. Beyond the identified stress response genes, a list of genes whose functions were previously not linked to genotoxic responses were further predicted in this study. These genes might functionally characterize a group of stress-response genes, which deserve further study. Moreover, we also envision that this integrative omics strategy will be applicable to discover O-GlcNAc functions and target genes in diverse physiological models.

In addition to the O-GlcNAc-regulated network, the results from motif enrichment and ChIP-seq signal distribution analyses implied that NRF1 is a master regulator that is necessary for complex transcriptome changes and maintains the homeostasis of cancer cells during exposure to genotoxic stress. NRF1 is considered a TF that plays vital roles in DNA synthesis and mitochondrial function[59]. The loss of NRF1 reduced tumor burden in a mouse model[60]. A large number of putative NRF1 targets, which were provoked by genotoxicity, were involved in our transcriptional regulatory network. NRF1 was found to interact with OGT and HCF-1 in our study. We further demonstrated that the O-GlcNAc modification of NRF1 enhances its stability and target gene expression. Our study of NRF1 ChIP-seq in two cell models clearly proves that O-GlcNAc promotes this TF assembly with chromatin and activates downstream gene expression during the response to genotoxic stress. Several representative targets, including *NSMCE2*, were found to be downregulated by NRF1 O-GlcNAc amino acid site mutations. In addition, the protective efficacy of ADR cells was significantly attenuated. These results also provide proof for an inference through which O-GlcNAc can enhance TF chromatin binding and transcriptional regulatory function. O-GlcNAc NRF1 can be viewed as a key response regulator in O-GlcNAc-modulated transcriptional reprogramming and genotoxic adaptation.

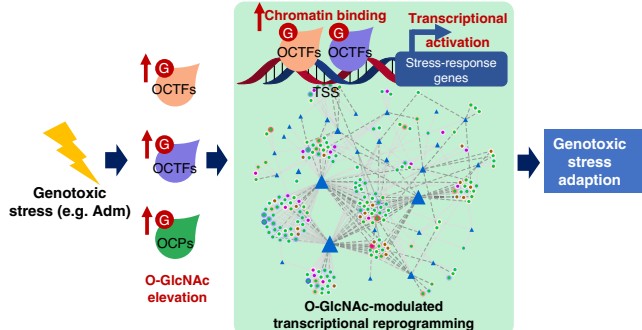

**Fig. 9 Proposed model for the genotoxic stress-responsive transcriptional reprogramming orchestrated by multiple OCTFs in breast cancer cells.** Genotoxicity provokes O-GlcNAc (G) elevation and dynamic changes in multiple OCTF genomic binding sites. The activity of multiple OCTFs, including NRF1, modulates a network of transcriptome upregulation to induce a holistic effect on cell fates in response to genotoxic stress.

In summary, we used the metabolic chemical reporter-based integrative omics strategy to provide the insights into the mechanisms of genotoxic stress-response transcriptional reprogramming orchestrated by the response regulator O-GlcNAc NRF1 and facilitated by other OCPs in breast cancer cells. We propose that the elevation in genomic binding of multiple OCPs could serve as a genotoxic stress sensor for the rapid elicitation of transcriptome alterations in cell fate decisions (Fig. 9). We also propose that the OCTF-regulated gene expression network plays an important role in promoting cancer cell resistance to chemotherapy. O-GlcNAc modification is a potential therapeutic intervention point for chemoresistance. This genome-wide chemical method is likely applicable to a broad range of O-GlcNAc modifications in the regulation of the progression and physiological processes of other diseases.

## Methods

**Cell culture and reagents.** MCF-7, MDA-MB-231, K562, HCT-15, and HEK 293T (293T) cells were obtained from Type Culture Collection of the Chinese Academy of Sciences (Shanghai, China) and were used within 6 months from resuscitation. All the cells were cultured in 90% RPMI-1640 (Gibco) supplemented with 1% penicillin/streptomycin antibiotics (Gibco) and 10% fetal bovine serum (Gibco). Adm (Adriamycin, Sigma) or Fu (Fluorouracil, Sigma) was added to cell cultures in stepwise increasing concentrations from 0.1 to 10 μM for eight months to develop genotoxicity-adapted variants, namely MCF-7/ADR, K562/ADR and HCT-15/FU, correspondingly. To maintain the genotoxicity-adapted phenotype, the complete medium of the variants was supplemented with 1 μM Adm or 150 μM Fu. MCF-7/ADR, K562/ADR and HCT-15/FU cells were maintained in complete medium without Adm or Fu for 1 week and cells with >90% viability before subsequent treatments. L01 was purchased from BioBioPha Co., Ltd. PugNAc and MG132 and cycloheximide (CHX) were purchased from Sigma. Other reagents were used as analytic grade or better.

**Plasmids, siRNA and transfection.** Full-length wild type (WT) human *NRF1* and O-GlcNAc amino acid sites mutant (human NRF1, Thr342/Thr500 → Ala) *NRF1* were subcloned into pCMV-Puro64. *NSMCE2* promoter report constructs were made by cloning 1046 bp of human *NSMCE2* promoter into luciferase vector pGL3-basic. The primers used for this study are listed in Supplementary Data 16. SP1 siRNA (#sc-29487), KLF5 siRNA (#sc-37718), NRF1 shRNA (#sc-38105-SH) and NSMCE2 siRNA (#sc-77813) were purchased from Santa Cruz Biotechnology. Transfection of the 293T, MCF-7 and MCF-7/ADR cells was performed with Lipofectamine 2000 (Invitrogen) according to the manufacturer's instructions. The stably transfected cells were then selected by the addition of puromycin (Sigma) to the medium.

**Chemoenzymatic labeling of O-GlcNAc NRF1.** To chemoenzymatic label of O-GlcNAc NRF1, the cells in six-well plates were lysed with western/IP lysis buffer (Beyotime, #P0013) supplemented with protease inhibitor cocktail (Roche) and the lysates were resuspended in the buffer containing 20 mM Hepes buffer (pH 7.9) containing 1% SDS with a final concentration of 1 mg/mL. O-GlcNAc proteins in

the lysates were labeled by GalNAz and biotin alkyne with the Click-iT™ O-GlcNAc Enzymatic Labeling System and the Click-iT™ Glycoprotein detection kit (Biotin alkyne) according to the manufacturer's instructions (Invitrogen). After labeling, the proteins were precipitated using the methanol/chloroform protocol and resuspended in 700 μL of enrichment buffer (1% Triton X-100 (v/v) and 0.1% SDS (w/v) in PBS) and precleared with 100 μL of vehicle-magnetic beads (BEAVER Life Science, # 70301). The streptavidin-magnetic beads (200 μL, BEAVER Life Science, #22308) was transferred into the above solution to capture the O-GlcNAc proteins (4 °C for 2 h on a rotating stand). The beads were collected, washed three times with the enrichment buffer, resuspended in SDS-PAGE loading buffer for the immunoblotting of NRF1. Experiments were independently repeated twice.

**Immunoblotting/lectin blotting and co-immunoprecipitation (co-IP).** For immunoblotting/lectin blotting, cells were harvested and lysed by western/IP lysis buffer supplemented with protease inhibitor cocktail (Roche) and phosphatase inhibitor cocktail (Roche) for 20 min at 4 °C. Total protein was quantified by Bradford protein quantification assay (Bio-Rad). Equal amounts of protein sample were mixed with SDS-PAGE loading buffer and boiled for 5 min. The proteins were resolved on SDS-polyacrylamide gel electrophoresis gels and transferred to poly-vinylidene fluoride membrane (Immobilon-P, Millipore-Sigma). After blocking with 5% non-fat dry milk for 1 h at room temperature, primary antibody was added at 4 °C and left overnight. After washing, the membrane was incubated with HRP-conjugated secondary antibodies for 1 h at room temperature. The primary antibodies used were anti-O-GlcNAc CTD110.6 (BioLegend, #838004, 1:1000), anti-O-GlcNAc RL2 (Abcam, #ab93858, 1:1000), anti-SP1 (Abcam, #ab13370, 1:1000), anti-KLF5 (CST, #51586, 1:1000), anti-NFKB1 (CST, #13586, 1:1000), anti-JUN (Proteintech, #24909-1-AP, 1:500), anti-NRF1 (CST, #46743, 1:1000), anti-OGT (Proteintech, #11576-2-AP, 1:1000), anti-GAPDH (CST, #5174, 1:1000), anti-HCF-1 (CST, #50708, 1:1000), anti-Flag (CST, #14793, 1:1000), anti-Histone 3 (CST, #4499, 1:1000), anti-NSMCE2 (Abcam, #ab241564, 1:1000), anti-Ubi (Abcam, #ab134953, 1:2000). Lectin sWGA (Vector Laboratories, #B-1025S, 1:2000) was used for lectin blotting. The appropriate secondary antibody used were anti-mouse IgG-HRP (CST, # 7076, 1:20,000), anti-rabbit IgG-HRP (CST, #7074, 1:20,000), anti-mouse IgM-HRP (Abcam, #ab97230, 1:20,000), Streptavidin-HRP (CST, #3999, 1:50,000), and the signals were detected by the ECL Plus kit (GE Health-care). All blots are representative of at least two independent experiments.

For co-IP, cells were harvested and lysed with western/IP lysis buffer supplemented with protease inhibitor cocktail (Roche) and phosphatase inhibitor cocktail (Roche) 20 min at 4 °C. The cell lysate was centrifuged for 10 min at 20,000 × g and 4 °C, and the supernatant was incubated with primary antibodies and protein A/G-magnetic beads (Bimake, #B23201) rotating at 4 °C overnight. Immunoprecipitates were washed five times with cold western/IP lysis buffer and then subjected to immunoblotting analysis. The antibodies and beads used for IP were anti-SP1 (Abcam, #ab13370, 1:100), anti-KLF5 (CST, #51586, 1:100), anti-JUN (Proteintech, #24909-1-AP, 1:50), anti-NRF1 (CST, #46743, 1:100), and anti-Flag-magnetic beads (Bimake, # B26102). To confirm O-GlcNAc modification of TFs and rule out the contamination of immunoprecipitates interacting with other non-O-GlcNAc proteins, cells were lysed with 1 × SDS lysis buffer (Beyotime, #P0013G) at 95 °C for 15 min. The denatured lysates were centrifuged and the supernatant was diluted with western/IP lysis (1:15) followed by IP with corresponding antibodies or anti-Flag-magnetic beads. The immunoprecipitates analyzed with anti-O-GlcNAc CTD110.6 antibody. Experiments were independently repeated at least twice.

**Chromatin complexes capture.** Cells ($1 \times 10^7$) were washed with PBS and then crosslinked with 1% formaldehyde for 15 min, and the crosslinking was quenched with 2.625 M glycine. The cells were scraped from the plate, centrifuged, and resuspended in 1 mL of hypotonic buffer (10 mM HEPES (pH 7.5), 10 mM KCl, 0.1 mM MgCl₂, 0.4% Igepal CA-630 (v/v), protease inhibitor cocktail (Roche) and phosphatase inhibitor cocktail (Roche)) until the cytomembrane was broken. The nuclei were centrifuged, washed four times in hypotonic buffer, homogenized in 500 μL of lysis buffer (50 mM HEPES (pH 7.5), 150 mM NaCl, 1.5 mM MgCl₂, 1% Igepal CA-630 (v/v), 0.1% SDS (w/v), protease inhibitor cocktail and phosphatase inhibitor cocktail) and incubated on a rotating stand for 1 h at 4 °C. The lysed nuclei were then sonicated with a Sonics Vcx130pb until the DNA was 200–500 bp and centrifuged at 20,000 × g at 4 °C for 15 min. The supernatant was collected as to obtain the crosslinked chromatin complexes. For the immunoblotting of the chromatin binding proteins, crosslinked chromatin complexes were treated with 5 μg/mL benzonase nuclease (HaiGene) and 10 μg/mL RNase (Sigma) and then subjected to the indicated protein analysis.

For the purification of sWGA-binding chromatin proteins, crosslinked chromatin proteins prepared as described above were decrosslinked in recovery buffer (20 mM Tris-HCl (pH 4), 2% SDS (w/v), 0.2 M glycine) at 100 °C for 20 min and then 60 °C for 2 h[28]. The decrosslinked chromatin proteins were diluted with western/IP lysis buffer (1:15), precleared with 100 μL of vehicle-agarose beads (Vector Laboratories), incubated with sWGA-agarose beads (Vector Laboratories, #AL-1023S), washed five times with western/IP lysis buffer, and then analyzed with the indicated antibodies.

**Cell viability assay**. Cells at a density of ~2000 cells/well were seeded in a 96-well plate and treated with vehicle or the indicated doses of Adm with/without L01 or PugNAc for 48 h. The viable cells were determined by Enhanced Cell Counting Kit-8 (Beyotime, #C0041) according to the manufacturer's instructions. The absorbance of each well at 450 nm was measured and normalized to that of the vehicle-treated wells. The viable cells were also stained with crystal violet (0.5% w/v) and imaged using a microscope. Results were reproduced in two biologically independent experiments.

**Quantitative real-time PCR analysis (qPCR)**. Total RNA was isolated using the Trizol method (Invitrogen). A total of 5 µg of RNA were reverse transcribed and amplified using One-Step SYBR Prime-Script PLUS RT-PCR Kit (TaKaRa) and the Thermal Cycler Dice instrument (TaKaRa) according to the manufacturer's instructions. The primers used in this study are listed in Supplementary Data 16. The relative enrichment of the different subunits at each site was calculated using the 2-ΔΔCt method. Experiments were performed in triplicate and repeated at least twice.

**Metabolic labeling and click reaction**. In 10-cm dishes, the cells at 30% confluence were incubated with culture medium containing 1 mM GalNAz (kindly provided by Prof. Xing Chen at Peking University) for 48 h. Approximately $5 \times 10^7$ cells were harvested by trypsin digestion and washed for three times with PBS. Crosslinked chromatin of the metabolically labeled cells was captured as mentioned above. To specifically biotinylated the O-GlcNAz-modified proteins, chromatin was subsequently incubated with 100 µM alkyne-biotin (Click Chemistry Tools, #1266-25), 100 µM BTTAA (Click Chemistry Tools, #1236-500), 50 µM $CuSO_4$ and 2.5 mM freshly prepared sodium ascorbate at 25 °C for 2 h. Ten milliliters methanol was then added to the above solution, and the mixture was stored at −80 °C for 4 h. The precipitants were centrifuged at $10,000 \times g$ at 4 °C for 15 min, washed twice with ice-cold methanol, and dissolved in 1 mL recovery buffer at 100 °C for 20 min and then at 60 °C for 2 h[28] to decrosslink the remote PPI and avoid the nonspecific enrichment of non-O-GlcNAz proteins. The resulting solution was diluted with western/IP lysis (1:15) and precleared with 100 µL of vehicle-magnetic beads (BEAVER Life Science, # 70301), and subsequently, 200 µL of streptavidin-magnetic beads (BEAVER Life Science, #22308) was transferred into the above solution to capture the OCPs. The resulting mixture was incubated at 4 °C for 4 h on a rotating stand. To reduce the contamination of nonspecific proteins during the above-described enrichment procedure, the beads were then strictly washed five times at 4 °C with low-salt buffer (composition 20 mM Tris-HCl (pH 8.1), 0.1% SDS, 2 mM EDTA, 1% TritonX100, 150 mM NaCl), five times with high-salt buffer prepared as described above but with 500 mM NaCl, and twice with LiCl buffer (10 mM Tris-HCl (pH 8.0), 0.25 M LiCl, 0.5% NP-40, 1% sodium deoxycholate, 1 mM EDTA). The resulting beads were resuspended in SDS-PAGE loading buffer for the immunoblotting of OCPs. For LC-MS/MS analysis, the beads were resuspended in 500 µL of 6 M urea in PBS. The released OCPs were reduced with 5 mM DTT at 80 °C for 10 min and alkylated with 10 mM iodoacetamide in the dark at room temperature for 30 min. After the buffer was exchanged with 200 µL of 2 M urea in PBS, 4 µL of trypsin (Promega, 0.5 µg/µL) and 2 µL 100 mM $CaCl_2$ were added to digest the OCPs. The resulting mixture was incubated at 37 °C for 16 h and then subjected to LC-MS/MS analysis.

**LC-MS/MS analysis and label-free quantification**. For whole-cell proteomics, ~$5 \times 10^6$ cells were washed three times with cold PBS and subsequently harvested using a protein lysis buffer containing 8 M urea, 400 mM ammonium bicarbonate and protease inhibitor cocktail. The cell lysis was improved by ultrasound. The sample was centrifuged for 10 min at 4 °C and 20,000 g, and the supernatant was used as the lysate. A total of 40 µg protein was trypsinized overnight at 37 °C. Nine biological replicates of the proteomic analysis were performed.

Whole-cell proteins and OCP peptides were redissolved in solvent A (0.1% FA in 2% ACN) and directly loaded onto a trap column (Acclaim PepMap 100, Thermo Fisher Scientific). Peptide separation was performed using a reversed-phase analytical column (5 µm C18, 75 µm × 25 cm, home-made) with a linear gradient of 5–26% solvent B (0.1% FA in 98% ACN) for 85 min, 26–35% solvent B for 10 min and 35–80% solvent B for 8 min at a constant flow rate of 1 µL/min on an UltiMate 3000 RSLCnano system. The resulting peptides were analyzed using a Q Exactive HF-X mass spectrometer (Thermo Fisher Scientific). The peptides were subjected to electrospray ionization (ESI) followed by MS/MS coupled online to UHPLC (Thermo Fisher Scientific). The intact peptides were detected with an Orbitrap at a resolution of 60,000. Peptides were selected for MS/MS using 28% NCE, and ion fragments were detected with the Orbitrap at a resolution of 15,000. A data-dependent acquisition procedure that alternated between one MS scan followed by 30 MS/MS scans was applied for the top 30 precursor ions above a threshold ion count of 3E6 in the MS survey scan with 30.0 s dynamic exclusion. Automatic gain control (AGC) was used to prevent overfilling of the ion trap; 1E5 ions were accumulated for the generation of MS/MS spectra. For the MS scans, the m/z scan range was 350 to 1500. The raw mass spectral data obtained in our study are available via iProX with the identifier PXD016713.

Protein identification and quantification were performed using MaxQuant with an integrated Andromeda search engine (version 1.5.3.28). The MS/MS data were searched against the Swiss-Prot human database (20,379 sequences) concatenated with the ReverseDecoy database and protein sequences of common contaminants. Trypsin was specified as a cleavage enzyme allowing up to two missing cleavages, five modifications per peptide and five charges. Carbamidomethylation on cystine was specified as fixed modification and oxidation on Met and acetylation on protein the N-terminus were specified as variable modifications. The FDR thresholds for protein, peptide and modification site were specified at 0.01. The minimum peptide length was set to 7. The other parameters in MaxQuant were set to the default values.

For the qualitative analysis of whole-cell proteins and OCPs, the representative proteins were identified in at least six of the nine replicates analyzed and only the proteins located in the nucleus were regarded as OCPs. For quantification of the label-free data (intensity ratio calculation), unique + razor peptides were used in the protein quantification with 2 minimum ratio counts based on the ion intensities of peak areas observed in the LC-MS spectra. Only those proteins present in six out of nine replicates, in at least one group were used for further statistical processing. The missing values were imputed from a normal distribution (downshift of 1.8 standard deviations and a width of 0.3 standard deviations). Proteins that met the expression fold change ≥ 2 for differential levels and p value ≤ 0.05 (two-sided unpaired Student's t-test) with multiple hypothesis testing correction using the Benjamini–Hochberg FDR cutoff of 0.05 were considered for the analyses. A principal component analysis and a Pearson correlation analysis with hierarchical clustering based on the Euclidean distance were performed to determine the reproducibility of the replicates. The annotation and GO functional enrichment analysis was performed using Metascape software (v3.0)[61]. STRING software (v11.0)[62] was used for the visualization of protein networks. Other bioinformatic analyses were performed using the phyper function in the R software package (v3.4.3), including beanplot (v1.2), ggplot2 (v3.0.0), igraph (v1.2.1), venneuler (v1.1-0).

**COGC-seq/ChIP-seq and bioinformatics**. Approximately $5 \times 10^7$ cells were used for COGC-seq. Crosslinked chromatin complexes were isolated from the GalNAz metabolic labeled cells and biotinylated as described above. Biotinylated O-GlcNAz-chromatin complexes were precleared and subsequently captured by incubating with 200 µL of streptavidin-magnetic beads at 4 °C for 4 h. The beads were strictly washed with low-salt, high-salt and LiCl wash buffer as for the OCP enrichment. The beads were resuspended in 100 µL TE buffer and digested with 10 µg/mL RNase (Sigma) at 37 °C for 30 min. The resulting solution was adjusted to 0.5% SDS and decrosslinked with 0.2 mg/mL Proteinase K at 65 °C overnight. DNA was purified with a MiniBEST DNA Fragment Purification Kit (Takara, #9761) and used for qPCR or sequencing. Two biological replicates of the sequencing were performed.

For sWGA lectin and NRF1 ChIP-seq, crosslinked chromatin complexes were captured from ~$3 \times 10^7$ cells and then sonicated with a Sonics Vcx130pb. The chromatin complexes were precleared with vehicle-agarose/magnetic beads and immunoprecipitated with sWGA-agarose beads, anti-Flag-magnetic beads or protein A/G-magnetic beads. The beads were washed and digested with the same buffer as that used for COGC-seq. The resulting chromatin complexes were decrosslinked using proteinase K. DNA was purified and used for qPCR or sequencing. sWGA lectin ChIP-seq was performed once.

ChIP followed by quantitative real-time PCR (ChIP-qPCR) was performed using Simple ChIP Kit (CST, # 9003) according to the manufacturer's instructions. Two micrograms of non-immune IgG (#sc-2027, Santa Cruz), anti-SP1 (Abcam, #ab13370, 1:10), anti-KLF5 (CST, #51586, 1:10), anti-NRF1 (CST, #46743, 1:10) and anti-Flag-magnetic beads (Bimake, 100 µL) were used for ChIP. Immunoprecipitated DNA was analyzed by qPCR. Non-immunized IgG and off-target primers (random primers that could not specifically bind the gene promoter region) was used as the negative control for validation of non-specific binding at the various binding sites. The primers for ChIP-qPCR used in this study are listed in Supplementary Data 16.

Next-generation sequencing libraries were generated and amplified for 15 cycles using a BGISEQ kit. The DNA fragments (100–300 bp) were gel-purified and sequenced with BGISEQ-500 (BGI). The raw COGC-seq and ChIP-seq data are available in the Gene Expression Omnibus database under the accession code GSE141698. The raw datasets were mapped to the hg19 human reference genome using Bowtie2[63] (v2.3.4.3) and Samtools (v1.2). No more than two mismatches were allowed in the alignment. Reads that were mapped only once at a given locus were allowed for peak calling. Enriched binding peaks were generated after filtering through control input using MACS2 (v2.1.1)[64]. MAnorm[33] (v1.2.0) and DiffBind (v2.10.0)[34] were used to normalize the mapped read counts for peak regions and identify differential quantitative regions (fold change ≥ 2 and P value ≤ $10^{-5}$ for MAnorm, FDR ≤ 0.05 for DiffBind). Bigwig files were generated using deepTools[65] (v3.3.2.0.0) and visualized in Integrative Genomics Viewer (IGV, v2.5.1). The genomic distribution, annotation comparison and visualization of O-GlcNAc binding sites were analyzed using Bedtools (v2.27.1) and ChIPseeker (v1.18.0)[66]. Motif discovery and enrichment analyses were performed using sequences from the peaks based on Homer[67] (v4.11) and the MEME suite[68] (v5.1.1). Heat maps of the sequencing signal density and sample Pearson correlation coefficients were generated by deepTools. The GO functional enrichment analysis was performed with Metascape[61]. The transcriptional reprogramming network was constructed

using Cytoscape software (v 3.6.1). Other bioinformatic analyses were performed using the phyper function in the R software package, including beanplot, ggplot2, igraph, venneuler.

**Luciferase assay**. 293T cells were plated at a density of $1 \times 10^4$ cells per well in 96-well plates and then cotransfected with 0.1 mg of *NSMCE2*-promoter-driven luciferase plasmids and 0.1 mg of WT-NRF1 or AA-NRF1 expression plasmids. The cells were treated with DMSO or 100 μM L01 for 48 h, and the reporter gene activities were measured with a dual luciferase assay system (Promega, #E1910) according to the manufacturer's instructions. The pGL3-basic vector was used as a control. Experiments were performed in triplicate biologically independent experiments and repeated twice.

**RNA sequencing**. Approximately $1 \times 10^4$ MCF-7 or ADR cells were collected and washed three times with ice-cold PBS. RNA extraction was performed using TRIzol reagent (Invitrogen). rRNA and genomic DNA were removed using a MICROBExpress Kit (Ambion) and DNase I (Invitrogen). The RNA was sheared, reverse transcribed with N6 random primers to obtain cDNA library and then qualified with an Agilent 2100 Bioanalyzer. Library sequencing was then performed using a combinatorial probe-anchor synthesis (cPAS)-based BGISEQ-500 sequencer (BGI) and base calling was conducted using BGISEQ-500 software (v0.3.8.1)[69].

The raw RNA sequencing data are available in the Gene Expression Omnibus database under the accession code GSE141698. Two biological replicates were used for RNA-seq. The raw reads were filtered to remove the adaptor reads. SOAPnuke software (v2.1.0)[70] was used to remove reads with more than 10% unknown bases and low-quality reads (the ratio of the bases with quality value $Q \leq 15 > 50\%$). Bowtie2 and HISAT2 (v2.1.0)[71] software were used to align the clean reads to the reference genome hg19. RSEM (v1.2.4)[72] was then used to quantify the gene expression level using the fragments per kilobase of transcript per million mapped reads (FPKM) method. The DEGs were determined by DESeq2[73] (v1.18.1, fold change ≥ 2 and Benjamini–Hochberg FDR ≤ 0.001). The GO functional enrichment analysis was performed using Metascape software. Other bioinformatic analyses were performed using the phyper function in the R software package, including beanplot, ggplot2, igraph, venneuler.

**Statistics and reproducibility**. Statistical analyses were performed with two-sided unpaired Student's *t*-test for single comparison using the GraphPad Prism 8.0 software and Microsoft excel 2019. *p* values < 0.05 was taken as statistically significant. The data are expressed as means ± SEM. To ensure reproducibility, blots and micrographs were repeated at least twice as indicated in the specific methods and legends. sWGA lectin ChIP-seq was performed once. Other RNA-seq, ChIP-seq and COGC-seq were repeated two times (biological replicates). Other experiments were performed at least two times (independent replicates with three biological replicates each). All results were successfully repeated. For Box plot: center-line is the median, box limits represent 25th and 75th percentile and whiskers are minimum and maximum. Detailed n values for each panel in the figures are stated in the corresponding legends.

**Reporting summary**. Further information on research design is available in the Nature Research Reporting Summary linked to this article.

## Data availability

The raw data of COGC-seq, NRF1 ChIP-seq and RNA-seq is available in the Gene Expression Omnibus database under the accession number GSE141698. The other published ChIP-seq and other sequencing raw data used in this study is available in the Gene Expression Omnibus database: MCF-7 cells ChIP-seq H3K27ac, H3K4me3[37] (GSE97481), H3K27me3[36] (GSE96363) and H3K4me1[36] (GSE86714), HCF-1[36] (GSE91992), SP1[36] (GSE92014), NRF1[36] (GSE91522), HEK293 cells OGT, O-GlcNAc ChIP-seq[74] (GSE36620), BJAB cells O-GlcNAc ChIP-seq[75] (GSE86154), MCF-7 cells RNA-Pol II ChIP-seq[76] (GSE34001) and MCF-7 GRO-seq[77] (GSE96859). The raw mass spectral data in our study is available via iProX with the identifier PXD016713. The raw data underlying Figures 1b–c, 2a, 4e, 5c, 5e–f, 6a, 6d–i, 7d, 8a–d, as well as Supplementary Figures 1a, b, d, 2, 3, 4, 6c, 7e, 9a, 11a, 12a, b, 13a–c, 14b, 15a, 16a, 17a–e, 19, 20, 21a–e are available in the Source Data file. All other data generated or analyzed during this study are available from the corresponding author on reasonable request. Source data are provided with this paper.

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

## Acknowledgements

We thank Prof. Xing Chen at Peking University for kindly providing GalNAz and helping with MS data analysis. This study is supported by the National Natural Science Foundation of China (31870793, 31971214), the National Science and Technology Major Project of China (2018ZX10302205), Natural Science Foundation of Liaoning Province (2019-MS-042) and the Fundamental Research Funds for the Central Universities (DUT20YG130, DUT20YG116).

## Author contributions

J.Z. and YB.L. conceived and designed the study. YB.L., N.Z., Y.C., and X.X. performed all labeling cell culture experiments and generated samples for LC-MS/MS and sequencing; Q.C., K.Z., and Y.R. performed LC-MS/MS; YB.L., N.Z., and W.L. prepared RNA samples and performed qPCR experiments; YB.L., T.D., K.L., YM.L., and X.H. performed all bioinformatics analyze. YB.L. and J.Z. analyzed data. YB.L. and J.Z. drafted the manuscript, while all authors provided input into the manuscript.

## Competing interests

The authors declare no competing interests.
