## [Peer Review File · Nature Communications]

Reviewers' comments:

Reviewer #1 (Remarks to the Author):

Zhang and coworkers performed three different profiling experiments with genomic, transcriptomic and proteomic technologies, aiming at resolving the links between genotoxic stress and O-GlcNAc modification on chromatin-associated transcriptional factors (TFs). The O-GlcNAc is an intracellular monosaccharide PTM that has been widely known to increase upon cell stress, though the molecular mechanisms are not clear. By integrating these rich datasets together, the authors discovered that many TFs are more heavily modified with O-GlcNAc when cells are challenged by genotoxic reagents, and the modified TFs tend to associate with genes that are functionally implicated with stress response and adaptation. They chose one TF, NRF1, for more detailed characterization, and showed that genetic and pharmacological perturbation of the O-GlcNAc modification level on NRF1 can regulate a large number of genes transcriptionally and rewire the sensitivity of cancer cells to genotoxic challenges. Overall, the work is with huge amount of data and a lot of omic-level analyses, while the detailed biochemical characterizations are relatively weak. Some concerns are listed:

1. What is the exact role of O-GlcNAc here? Stabilize target proteins or increase the binding of TFs with corresponding genomic loci? Or both? I could not get a clear-cut conclusion from this study. Take NRF1 as an example, do the WT and mutant proteins bind to chromatin with similar or different strengths? Does O-GlcNAc affect the stability of NRF1 only or the PTM can destabilize the NRF1's association with chromatin?
2. The authors used an unnatural sugar reporter probe to label proteins with O-GlcNAc and pulled out these modified proteins for proteomic analysis. The pipeline has been demonstrated effective to profile substates of O-GlcNAc modifications. However, it was clear to me that how many replicates of proteomic experiments have been performed as the label free quantification is known to be prone to high stochasticity and perturbation.
3. As the authors created an artificial cell model with genotoxic challenges, they should at least demonstrated that the global landscapes of normal and adapted cells do not differ significantly by quantitative whole-cell proteomics, as otherwise it is less meaningful to compare subproteomes with specific PTMs.
4. The formaldehyde crosslinking can capture protein-protein interactions. As the authors decrosslinked the proteins from chromatin, the samples will be contaminated with other secondary

and even more remote interacting proteins from the actual chromatin-TF interfaces. Please evaluate and comment on how these contaminations will affect the analysis and conclusions.

5. The manuscript was written very poorly with many places of super long sentences that are hard to follow, and there are so many grammatical errors that should be cleaned up thoroughly. Just to name a few, "stress was exisited", "illustrate this speculate", "represented approximately 30-fold the IC50", "would be merged' ...

Reviewer #2 (Remarks to the Author):

The manuscript "Proteomic profiling and genome-wide mapping of O-GlcNAcylated chromatin-associated proteins reveals O-GlcNAc-regulated genotoxic stress response in human breast cancer cells" provides a systems biology approach to understanding the response of human breast cancer cells to adriamycin. Robust methods were used to cross check the observations. The authors have taken some care to avoid the complications of metabolic labelling, namely the off-target effects of donor sugars that are peracetylated. Although numerous transcriptional networks were identified, the authors have focused on the NRF1 response for detailed analysis.

Issues emerging in review:

1. This is a sophisticated analysis involving multiple approaches making integration difficult. A simple statement of how the integration was performed would be helpful.

2. It is not clear why NRF1 was chosen over all of the other key networks identified.

3. The motif analysis of enriched binding sites sheds little light on the actual cis elements targeted since co-occupancy by many transcription factors is likely.

In summary, this is a very complete paper. The regulation of NRF1 by O-GlcNAc has been reported, reducing the novelty of the observations, but the completeness of this study is noteworthy.

Reviewer #3 (Remarks to the Author):

The manuscript by Liu et al. describes an integrated chemical and sequencing strategy to investigate how increased O-GlcNAc modification may protect cells against stress by promoting the transcription of key genes. To accomplish this, they used GalNAz labeling in combination with chromatin precipitation to identify potentially O-GlcNAcylated transcription factors across the genome. They then generated different cancer cell lines that are resistant to the cell stress agent Adriamycin and showed that they display an increase in O-GlcNAc modification that is required for their survival under stress. Next, they used a combination of RNA sequencing and GalNAz to identify transcriptional programs that might be controlled by increased O-GlcNAcylation. They then use small molecule inhibitors to show that global modulation of O-GlcNAc levels does indeed cause the expected up and down regulation of some of these identified transcriptional programs. Finally, they focus on one of these pathways, through NRF1. They find that two previously identified O-GlcNAc sites on this protein control its protein-chromatin and protein-protein interactions and transcriptional activity. This is a very thorough and well executed study. I think it will make an excellent addition to Nature Communications.

Reviewer #4 (Remarks to the Author):

Major points:

The authors have an interesting data set, but it remains largely descriptive with a lack of complementary rescue experiments and knock out experiments to validate their key findings. Additionally, the bioinformatic analysis lacks a high level of rigor or sophistication, making the reader seriously doubt the otherwise interesting findings herein.

The RNA-seq analysis lacks some routine and helpful tasks, such as GSEA (Gene Set Enrichment Analysis) to show a more robust documentation of the findings. This could be done with both public MSigDB gene sets, as well as gene sets defined by the COGC-seq data.

The shift from a balanced intronic/intergenic/promoter distribution of COGC-seq to a promoter-heavy distribution after Adriamycin resistance is fascinating, but not elucidated. This would indicate that this modification is present at enhancer elements (and, super enhancers?) in a baseline scenario, but then upon Adriamycin resistance the enhancers lose factors with this modification. To distinguish an absolute “loss of signal at enhancers” from a “gain of signal at promoters” would need exogenous chromatin spiked into the precipitation, with a species orthogonal antibody (try <https://www.activemotif.com/catalog/1091/chip-normalization>). Then, data should be normalized to this spiked in reference, to distinguish these alternate possibilities. The authors also need to do a better job by overlapping H3K27ac, H3Kme4, and repressive marks such as H3K27me3, so that the

“kind of chromatin” where this modification is found can be clearly seen (and in MCF7 much of this data is publicly available). This should include metagene plots not only of TSS locations, but also of other chromatin elements. And, locations of gain and loss, dividing into enhancers and promoters (and other chromatin types this modification could sit in), followed by metagene/heatmap plotting of the data, should be included to help interpret it more fully.

Lastly, NRF1 ChIP-seq would be expected in such a study, under the appropriate conditions, and with the proper cross-analysis, to prove the major claims that this factor is an essential communicator of this modification at genes of interest. The authors need to tell their story in light of this seminal manuscript: Domcke, S., et al 2015. Competition between DNA methylation and transcription factors determines binding of NRF1. *Nature*, 528(7583), pp.575-579.

General comments:

The paper is hindered greatly by the incorrect grammar throughout. Before this is acceptable for publication in *Nature Communications*, this must be cleaned up to make it readable. Or, resubmit to *Nature Miscommunications*. This reviewer has found an error in almost every sentence, but I will not correct all of them.

Specific remarks:

To improve the interpretability of “ADR”, in the figures I recommend having the full definition of this term presented upfront, perhaps with a schematic illustration of the several months of induced Adriamycin resistance achieved by the authors. (and, how many months specifically? 3 months? 14 months?)

Line 61 – edit to “levels in a stress”...

Line 71 – this sentence is incorrect grammar “Recently, chromatin O-GlcNAc modification is involved in gene expression”. It is missing a phrase such as “it has been discovered that chromatin O-GlcNAc...”

Line 103 – delete the incorrectly placed word “in”

Figure 1a: It would be easier to appreciate the method if azide (N₃) was in bold in the chemical structure.

Figure 2c: If O-GlcNAz is indiscriminately distributed to all transcribed genes, this plot would be expected to mirror RNA-seq (or, better, proteome-wide data without an O-GlcNAz pull down). The authors need to repeat this volcano plot as a series of volcano plots, where each plot has the gene from different RNA-seq Log₂FC bins (to see how much of this difference can be a simple reflection of expression differences). I might recommend binning each gene in the volcano plot into “Up in MCF7”, “Not differentially expressed” between MCF7 and ADR, and “Up in ADR”, using a threshold of ~Log₂FC of 3 for TPM RNA-seq values.

Figure 3a: The ADR vs. MCF7 motif comparison is rather weak on a few fronts. First, the GGGnGGG motifs for ADR are not different essentially from the CCCnCCC motifs found in MCF7 – they are just the mirror image of one another. Presumably a P-value of enrichment for each motif is available for

each cell line – why not, with each top ranking motif, share the motif enrichment across both cell lines?

Fig 4c: The legend has triangles “targeting less genes” twice.

Figure 3D/E – the authors need to perform GO enrichment analysis, perhaps using GREAT (<http://great.stanford.edu/public/html/>) or something similar, on their peaks (maybe on the overlapping peaks, and also on the non-overlapping peaks from each cell line). Something like that would be needed to justify the “leap” into genotoxic stress, over other possible gene pathway categories this mark might decorate.

Figure 6 – What does AA mean in “Flag-AA-NRF1”? Can you define this in the legend?

Response to Referees

Reviewer 1#

- 1. What is the exact role of O-GlcNAc here? Stabilize target proteins or increase the binding of TFs with corresponding genomic loci? Or both? I could not get a clear-cut conclusion from this study. Take NRF1 as an example, do the WT and mutant proteins bind to chromatin with similar or different strengths? Does O-GlcNAc affect the stability of NRF1 only or the PTM can destabilize the NRF1's association with chromatin?**

Reply

This reviewer expressed concerns regarding the exact role of O-GlcNAc in this study. To clarify this question, additional whole-cell proteomics and NRF-1 ChIP-seq assays were performed and are described in the revised manuscripts.

In fact, we provide evidence showing that genotoxic stress induces a striking increase in chromatin O-GlcNAc and then affects the genome loci of O-GlcNAc chromatin-associated proteins (OCPs), resulting in the reprogramming of gene expression in human breast cancer cells. Previous protein level studies revealed that, O-GlcNAc regulates protein stability. In the present study, several OCPs exhibited the same varying tendency in whole-cell proteomics (MCF-7 and ADR cells without O-GlcNAz enrichment) and GalNAz-labeled OCP sub-proteomes. We hypothesize that O-GlcNAc could regulate the stability and might further influence the chromatin binding of these OCPs. However, most other OCPs, which present the similar mRNA and whole-cell protein levels (Supplementary Figs. S7A and B), exhibit increased interaction with chromatin in ADR cells compared with MCF-7 cells (Fig. 2E). Consistent with the higher number of OCPs found in ADR cells, an increase in the COGC-seq peak height was observed in ADR cells compared with MCF-7 cells (Fig. 2G). A low signal of histone H3 trimethylated at lysine 27 (H3K27me3, marker of transcriptional repression) was found throughout O-GlcNAz sites in both MCF-7 and ADR cells, whereas a high signal of histone H3 acetylated at lysine 27 (H3K27ac, marker of transcriptional activation) was measured in all these regions, which

suggested that the binding of OCTFs is associated with transcriptional activation (Fig. 3D). Consistently, the majority of sufficiently expressed genes were occupied by OCTFs in both MCF-7 and ADR cells (Supplementary Fig. S16). Therefore, we conclude that O-GlcNAc can directly enhance the chromatin binding of these OCPs without changing their protein stability. OCTFs are associated with transcriptional activation in response to genotoxic stress.

Taking NRF1 as an example, we found that O-GlcNAc enhances the stability of NRF1 and promotes its dynamic assembly with chromatin during the genotoxic stress response (Fig. 6H and Supplementary Fig. S20). Notably, even though the two types of recombinant NRF1 exhibited similar expression levels in stably transfected ADR cells, AA-NRF1 showed significantly less chromatin binding than WT-NRF1 (Fig. 6I), which indicated that the abundance of chromatin-bound NRF1 is monitored by O-GlcNAc modification. Consistently, the increase of in NRF1 binding was also accompanied by a significant increase in the expression of downstream genes (Figs. 7D and 6B). Furthermore, the comparison of WT-NRF1 and AA-NRF1 ChIP-seq peaks in MCF-7 and ADR cells revealed that almost all the peaks (particularly in the TSS regions) showed a decreased NRF1-binding affinity after O-GlcNAc amino acid site mutation (Fig. 7E and Supplementary Fig. S22). O-GlcNAc inhibition and downregulation of NRF1 also suppressed these genes and protected against Adm stress in ADR cells (Supplementary Fig. S21), which suggested that O-GlcNAc can enhance the association of NRF1 with chromatin, and a pool of novel candidate genes could be upregulated by the O-GlcNAc modification of NRF1.

Together, our results show that O-GlcNAc indeed regulates the stability of several OCPs and can further influence their chromatin binding of them, resulting in the dynamic activation of diverse target genes during the response to genotoxic stress. We also conclude that O-GlcNAc can directly enhance the chromatin binding of other OCPs, including NRF1, and is essential for the transcriptional activation of many genes to maintain protection against genotoxic stimuli. We have added a discussion of the role of O-GlcNAc and revised the related text in the revised manuscript.

- 2. The authors used an unnatural sugar reporter probe to label proteins with O-GlcNAc and pulled out these modified proteins for proteomic analysis. The pipeline has been demonstrated effective to profile substrates of O-GlcNAc modifications. However, it was clear to me that how many replicates of proteomic experiments have been performed as the label free quantification is known to be prone to high stochasticity and perturbation.**

Reply

This reviewer was concerned about the number of replicates used in the proteomic experiments. Three biological replicates of the proteomic analysis were shown in the original manuscript. To eliminate the stochasticity and perturbation of label-free quantification, we performed nine biological replicates, and all these data are provided in the revised manuscript. We have also improved the threshold for data filtering. Protein must be identified with high confidence (in six out of nine replicates of at least one group). After valid value filtering, we performed further analysis of 1403 proteins. The heat maps shown in Supplementary Fig. S6A exhibits the high correlation among the different biological replicates of both cells. Subsequently, the Log_2 ratios of all the protein intensities were used to generate a heat map by hierarchical clustering. The heat map also showed the high correlation among the biological replicates. A PCA plot shows clear clustering between the MCF-7 versus ADR replicate samples (Supplementary Figs. S6B-C). Subsequent statistical analyses showed that 875 OCPs (458 located in the nucleus), including 88 TFs and cofactors, exhibited ≥ 2 -fold differences (p value ≤ 0.05 , FDR ≤ 0.05 , Fig. 2E). Only the candidate proteins, which presented nuclear localization, are regarded as the candidate differential quantitative OCPs. We have revised the related text in the new manuscript.

- 3. As the authors created an artificial cell model with genotoxic challenges, they should at least demonstrated that the global landscapes of normal and adapted cells do not differ significantly by quantitative whole-cell proteomics, as otherwise it is less meaningful to compare subproteomes with specific**

PTMs.

Reply

This reviewer suggested that we should additionally perform a quantitative whole-cell proteomics analysis of MCF-7 cells and genotoxicity-adapted ADR cells. We agree with this point and performed additional experiments (Supplementary Figs. S6D-F). The quantitative proteomics analyses of whole-cell proteins in MCF-7 and ADR cells identified 5145 proteins with high confidence (identified at least six times in nine biological replicates). Proteins were considered significantly deregulated if the following criteria were met: fold change ≥ 2 , p value ≤ 0.05 (multiple t-test) and FDR ≤ 0.05 . A total of 1219 proteins (including 229 OCPs) and 1176 proteins (including 120 OCPs) were upregulated and downregulated, respectively, in ADR cells compared with MCF-7 cells. As predicted, the protein expression levels obtained from the whole-cell proteomics analysis (MCF-7 and ADR cells without O-GlcNAz enrichment) and transcriptomic data showed a positive correlation in MCF-7 and ADR cells (Supplementary Fig. S6F). Although a limited number (349) of OCPs exhibited the same varying tendency in the whole-cell proteomics data and GalNAz-labeled OCP subproteomes (Supplementary Fig. S6E), the majority of OCPs (526 of 876 OCPs and 58 of 88 OCTFs) could not reflect the protein expression differences in the whole-cell proteomics without O-GlcNAz enrichment (Supplementary Fig. S7B), which indicated that O-GlcNAc can influence the chromatin binding of OCPs, independent of the protein expression levels. These results also suggest the specificity of this chemical reporter-based OCP enrichment. We have revised the text in the new manuscript.

4. The formaldehyde crosslinking can capture protein-protein interactions. As the authors decrosslinked the proteins from chromatin, the samples will be contaminated with other secondary and even more remote interacting proteins from the actual chromatin-TF interfaces. Please evaluate and comment on how these contaminations will affect the analysis and

conclusions.

Reply

This reviewer is also concerned about the nonspecific contamination of formaldehyde crosslinking in the profiling of the GalNAz-labeled OCP subproteomes. We realized that formaldehyde crosslinking would capture the protein-protein interactions (PPIs) between OCPs and other secondary or remote proteins. To eliminate the risk of these nonspecific contaminations, before the enrichment of biotinylated O-GlcNAz-modified proteins, the crosslinking PPI complex was decrosslinked (Supplementary Fig. S1A) using a previously reported extensively used method (Fowler, Carol B., et al., *Lab. Invest.* 2007, 87, 836) which was demonstrated to be effective in reversing the formaldehyde crosslinking. The O-GlcNAz PPI complex was dissolved in 1 mL of recovery buffer (20 mM Tris-HCl (pH 4), 2% SDS and 0.2 M glycine) at 100°C for 20 min and then at 60°C for 2 h. The OCPs were then enriched with streptavidin beads and strictly washed with standard low-salt and LiCl buffer used in ChIP. The resulting proteins were subjected to immunoblotting, sWGA lectin ChIP-seq or liquid chromatography-tandem mass spectrometry (LC-MS/MS) based proteomics analysis. The effectiveness of this method is shown in Supplementary Fig. S1A. Only the differential quantitative OCPs located in the nucleus were identified as regulatory candidates, further reducing the effect of nonspecific contamination in OCP capture step on the ultimate network construction and analysis. The specificity of this chemical-reporter-based OCP enrichment was also certified by the distinct difference between the OCP quantity and gene expression obtained by RNA-seq and whole-cell proteomics. In the revised manuscript, we added clear descriptions of the procedure used in this study and added a discussion of this issue.

- 5. The manuscript was written very poorly with many places of super long sentences that are hard to follow, and there are so many grammatical errors that should be cleaned up thoroughly. Just to name a few, “stress was**

existed”, “illustrate this speculate”, “represented approximately 30-fold the IC50”, “would be merged’ ...

Reply

We accept this reviewer’s comment and the English language in revised manuscript has been checked by *American Journal Experts*.

Reviewer 2#

- 1. This is a sophisticated analysis involving multiple approaches making integration difficult. A simple statement of how the integration was performed would be helpful.**

Reply

We agree with this reviewer’s comment and have explained how the integrative omics analysis was performed in the revised manuscript as follows: “The comparison of the RNA-seq DEGs (7112 genes) with differential COGC-seq peak-associated genes (2194 genes) yielded 976 overlapping genes (Supplementary Fig. S15). We subsequently linked these genes to the abovementioned OCTF-targeting genes (1550 genes) and found that 647 genes were directly regulated by 33 diverse OCTFs during adaptation to genotoxic stress (Supplementary Table S14). For the first time, we established an O-GlcNAc-regulated stress response gene expression network. (Fig. 5B).” We also revised the schematic of the integrative analysis in the new manuscript (Fig. 5A).

- 2. It is not clear why NRF1 was chosen over all of the other key networks identified.**

Reply

This reviewer questioned why we selected NRF1 as the key regulator in the

O-GlcNAc modulated gene expression network. The results from the revised motif enrichment and ChIP-seq signal distribution analysis implied that NRF1 is a master regulator that is necessary for complex transcriptome changes in the genotoxic stress response.

First, OCTFs were ranked based on their motif enrichment values (P value) and the number of genome-wide binding sites obtained by COGC-seq (Fig. 6A). Although the three most enriched central OCTFs (SP1, KLF5 and NRF1) were identified (Figs. 5B and 6A), little difference in the enrichment, binding sites and target genes of SP1 and KLF5 was observed between MCF-7 and ADR-biased COGC-seq peaks (we refer to peaks that persisted between MCF-7 and ADR as “unbiased”, and differential quantitative peaks that were either lost or gained during genotoxic stress are referred to as “MCF-7-biased” or “ADR-biased”, Fig. 3C). In contrast, substantial changes in the NRF1 chromatin-binding parameter in MCF-7- and ADR-biased peaks were detected. Consistent with this finding, 9.4% of the MCF-7 COGC-seq peaks were occupied by NRF1, whereas 45.6% of the ADR peaks overlapped with the NRF1 peaks (Fig. 6C), which suggested the influential role of this TF in the O-GlcNAc-regulated genotoxic stress response.

Second, we performed additional ChIP-seq signal distribution analysis of these candidate OCTFs. Because SP1 and KLF5 belong to the same TF family and bind to similar DNA motifs, the SP1 ChIP-seq signal was further analyzed. The SP1 ChIP-seq signal showed similar levels across MCF-7- and ADR-biased COGC-seq peaks (Supplementary Fig. S18A), which indicated that the chromatin-binding affinity of SP1 and KLF5 does not change between MCF-7 and ADR cells. SP1 and KLF5 cannot be the response factors during the genotoxic response. However, substantial changes in the NRF1 ChIP-seq signal distribution in MCF-7- and ADR-biased peaks were detected (Fig. 6B and Supplementary Fig. S18B). Consistent with this, 9.4% of the MCF-7 COGC-seq peaks were occupied by NRF1, whereas 45.6% of the ADR peaks overlapped with the NRF1 peaks (Fig. 6C), which suggested that the influential role of this TF in the O-GlcNAc-regulated genotoxic stress response.

Additional NRF1 ChIP-seq using two cell models (Figs. 7A-C) also clearly

proved that genotoxicity-induced O-GlcNAc enhanced the transcriptional regulatory function of NRF1. Several representative novel targets, including *NSMCE2*, were found to be downregulated by NRF1 O-GlcNAc amino acid site mutations. These results also provide proof for an inference through which O-GlcNAc can regulate TF chromatin binding and transcriptional regulatory function in response to genotoxicity. Therefore, O-GlcNAc NRF1 can be viewed as a key response regulator in O-GlcNAc-modulated transcriptional reprogramming. We added clear descriptions of these issues in the revised manuscript and also modified some of the text and certain figures.

3. The motif analysis of enriched binding sites sheds little light on the actual cis elements targeted since co-occupancy by many transcription factors is likely.

Reply

This reviewer was concerned about the availability of the predicted TF binding sites in the motif analysis. In fact, we were aware that the actual chromatin elements might be cooccupied by multiple TFs. If the TF-binding site analysis of COGC-seq peaks completely relied on computational methods, false enrichment of TFs that do not actually interact with chromatin might occur. In the revised manuscript, the scanning of MCF-7 and ADR-biased COGC-seq peaks using motif discovery algorithms (Homer) was preformed, and we found that diversified TF-binding sites were enriched in MCF-7- and ADR-biased COGC-seq peaks (Fig. 4A). Further motif enrichment analysis revealed that a vast majority of the peaks indeed contained multiple TF binding sites (Fig. 4B). An appreciable number of 367 putative TFs were identified in MCF-7 and ADR-biased COGC-seq peaks. To rule out the false enrichment of nonchromatin binding TFs, these 367 putative TFs overlapped with 88 differential quantitative OCTFs identified from proteomics datasets (Fig. 3C). Thirty-three candidate OCTFs that actually interacted with chromatin and were found to be enriched in the motif analysis were ultimately used for gene expression network construction. The chromatin binding of certain OCTFs was validated by CHIP-qPCR

(Supplementary Fig. S12). We added clear descriptions of these issues and revised the figures in the new manuscript.

Reviewer 4#

Major points:

- 1. The authors have an interesting data set, but it remains largely descriptive with a lack of complementary rescue experiments and knock out experiments to validate their key findings. Additionally, the bioinformatic analysis lacks a high level of rigor or sophistication, making the reader seriously doubt the otherwise interesting findings herein.**

Reply

This reviewer suggested that we perform additional rescue and knock out experiments to validate our key findings. We agree with this point, and performed additional experiments. Because SP1, KLF5 and NRF1 were identified as the most enriched central OCTFs (O-GlcNAc chromatin-associated TFs and cofactors identified by LC-MS/MS) in the O-GlcNAc-regulated gene expression network, we knocked down the expression of SP1 and KLF5 using siRNA in ADR cells. The transcription of the downstream genes *THUMPD3*, *OTUD7B*, *MAN2C1*, *SEC13*, and *PPP2R5B* and the viability of ADR cells under genotoxic stress were significantly decreased following the loss of these two important OCTFs, which suggested that these genes are directly regulated by O-GlcNAc SP1 and KLF5 during the genotoxic stress response (Supplementary Fig. S17). Furthermore, the stable knockdown of NRF1 using shRNA significantly increased the genotoxicity of Adm in ADR cells. Cell death could be reversed by rescue expression of WT-NRF1 but not AA-NRF1, which indicated the key role of O-GlcNAc NRF1 in genotoxic adaptation (Supplementary Figs. S21A and B).

To demonstrate the role of O-GlcNAc in stress-induced gene transcription and the cellular phenotype of genotoxic adaptation, we also conducted an analysis of

downstream gene mRNAs and cell viability in the presence of the O-GlcNAc inhibitor **L01** or agonist PugNAc. The transcription of the abovementioned OCTF-targeting genes was attenuated in ADR cells under **L01** treatment, whereas the expression of these genes was accumulated in PugNAc-stimulated MCF-7 cells (Fig. 5E). Moreover, **L01** significantly decreased the viability of ADR cells upon genotoxic provocation, whereas PugNAc revealed a reduction in the cytotoxicity of Adm in MCF-7 cells. We also performed additional experiments to compare the WT-NRF1 and AA-NRF1 ChIP-seq peaks in ADR cells. The results revealed that almost all peaks showed a decreased NRF1-binding affinity after O-GlcNAc amino acid site mutation (Fig. 7E and Supplementary Fig. S22B), which suggested that O-GlcNAc can enhance the association of NRF1 with chromatin. We added clear descriptions of these issues and revised the figures in the new manuscript.

This reviewer also suggested that we increase the sophistication of the bioinformatics analysis. We agree with this point, and additional bioinformatics analyses were conducted and have been incorporated in the revised manuscript. According to the quantitative comparisons of COGC-seq peaks using MAnorm, we identified 2198 MCF-7-biased and 1169 ADR-biased COGC-seq peaks (Fig. 3C). Based on these peaks, we revealed that OCTFs undergo an enhancer-promoter binding switch and dynamically associate with diverse target genes during the genotoxic stress response. We also performed additional gene clustering analysis and describe the NRF1 ChIP-seq signal distributions in the revised manuscript.

2. The RNA-seq analysis lacks some routine and helpful tasks, such as GSEA (Gene Set Enrichment Analysis) to show a more robust documentation of the findings. This could be done with both public MSigDB gene sets, as well as gene sets defined by the COGC-seq data.

Reply

This reviewer suggested that we perform a GSEA of the RNA-seq DEGs and COGC-seq-associated genes. We agree with this point, and the additional analysis is

shown in the revised manuscript. The GSEA revealed that ADR cells expressed genes involved in cellular stress and the DNA damage response at higher levels than the MCF-7 cells (Fig. 2C), which indicated that chromatin O-GlcNAc fluctuations regulate the expression of a large scale of genes. Additional GO analysis revealed that genes associated with ADR-biased COGC-seq peaks were more enriched for roles in the stress response than those of MCF-7-biased COGC-seq peaks (Fig. 4E). The GSEA also showed that OCTF-targeting genes in the O-GlcNAc-regulated gene expression network are enriched in pathways related to the stress response. GSEA of 976 overlapping genes identified by comparing RNA-seq DEGs with differential COGC-seq peak-associated genes showed enrichment in pathways related to the cellular stress response and chromatin conformational change. Together, these data suggest that OCTFs regulate the transcription of a large scale of genes related to the stress response to maintain homeostasis.

3. The shift from a balanced intronic/intergenic/promoter distribution of COGC-seq to a promoter-heavy distribution after Adriamycin resistance is fascinating, but not elucidated. This would indicate that this modification is present at enhancer elements (and, super enhancers?) in a baseline scenario, but then upon Adriamycin resistance the enhancers lose factors with this modification. To distinguish an absolute “loss of signal at enhancers” from a “gain of signal at promoters” would need exogenous chromatin spiked into the precipitation, with a species orthogonal antibody (try <https://www.activemotif.com/catalog/1091/chip-normalization>). Then, data should be normalized to this spiked in reference, to distinguish these alternate possibilities. The authors also need to do a better job by overlapping H3K27ac, H3Kme4, and repressive marks such as H3K27me3, so that the “kind of chromatin” where this modification is found can be clearly seen (and in MCF7 much of this data is publicly available). This should include metagene plots not only of TSS locations, but also of other chromatin elements. And, locations of gain and loss, dividing into enhancers and

promoters (and other chromatin types this modification could sit in), followed by metagene/heatmap plotting of the data, should be included to help interpret it more fully.

Reply

This reviewer suggested that we further investigate the shift in the COGC-seq signal distribution between enhancers and promoters. We agree with this comment and performed additional bioinformatics analyses.

In the original manuscript, a higher proportion of COGC-seq peaks located in the gene promoter regions were found in ADR cells (88.5%) than in MCF-7 cells (54.6%, Fig. 3A). As described in the revised manuscript, a signal distribution analysis revealed a clear shift in the COGC-seq signal from a promoter and intron/intergenic balanced distribution (MCF-7) to a promoter-biased distribution (ADR) during the genotoxic stress response (Fig. 3B). These results indicate that O-GlcNAc occupied enhancer elements in parental MCF-7 cells, but upon genotoxic adaptation, the enhancers lost TFs with O-GlcNAc.

To further elucidate how O-GlcNAz sites change during the course of the genotoxic stress response, we should examine the dynamics of COGC-seq peaks. Comparisons between MCF-7 and ADR COGC-seq signals can provide novel insights into differences in the OCTF chromatin element occupancy. Although a “spike-in” analysis should aid the identification of the global changes in OCTF binding, its applicability is limited in COGC-seq. As a prerequisite, only if the foreign reference genome applied in COGC-seq is labeled with azide will the subsequent chemoselective enrichment procedures proceed. Moreover, experimental variation and discordance of species-related biorthogonal specificity might also be introduced in COGC-seq, which would affect subsequent data interpretation. Therefore, two statistical ChIP-seq normalization approaches, MA_{norm} and DiffBind, which have been extensively used in recent studies, were employed to detect the genomic regions showing differential quantitative O-GlcNAc chromatin-associated protein (OCP) binding. The genotoxicity-induced differential binding regions determined by

MANorm and DiffBind were of a similar magnitude and highly coincident, reflecting the reliability of the results (Supplementary Fig. S10). Although a large fraction of peaks persisted between MCF-7 and ADR cells (we refer to such peaks as “unbiased”), we identified 3367 differential quantitative peaks that were either lost or gained during exposure to genotoxic stress (“MCF-7-biased” or “ADR-biased”, Fig. 3C) compared with the baseline scenario. Using these data, we highlight a genotoxicity-induced enhancer-promoter switch in OCTF genome binding sites.

To confirm the switch in the COGC-seq distribution between enhancers and promoters, we measured the distribution of published histone marker ChIP-seq signals at genotoxic stress-biased COGC-seq peaks. Overall, we found that COGC-seq datasets shared a high degree of conservation with transcriptionally activated chromatin (Supplementary Fig. S11A). A low H3K27me3 (marker of repression) signal was found throughout O-GlcNAz sites in both MCF-7 and ADR cells, whereas a high H3K27ac (marker of activation) signal was measured in all these regions, which suggested that the binding of OCTFs is associated with transcriptional activation (Fig. 3D). MCF-7-biased regions were surrounded by the highest levels of H3K4me1 (marker of enhancer) and lowest levels of H3K4me3 (marker of promoter), which suggested that OCTFs that are unique to MCF-7 cells play a predominant regulatory role at enhancers as opposed to promoters. In contrast, unbiased and ADR-biased sites showed distinct increases in the H3K4me3 level, whereas the H3K4me1 level was reduced in these sites. These patterns were also observed in analyses of the promoter and intron/intergenic regions (Supplementary Fig. S11B). Furthermore, genotoxicity reduced O-GlcNAz sites associated with reported enhancer and super-enhancer elements in MCF-7 cells (Fig. 3E and F). COGC-seq signals also decreased at these elements in ADR cells compared with MCF-7 cells (Figs. 3G and H). The results presented here support a regulatory mechanism through which O-GlcNAc and TFs coordinate enhancer-promoter switching to reprogram the expression of genes needed for the maintenance of cellular homeostasis. The shift from a balanced promoter/enhancer pattern of O-GlcNAz sites to a promoter-heavy distribution after genotoxic adaptation also provides evidence showing that

O-GlcNAc might also play a role in the chromatin conformation and occupancy of other regulators. We added clear descriptions of these issues in the revised results and discussion sections. Certain figures have also been revised.

4. Lastly, NRF1 ChIP-seq would be expected in such a study, under the appropriate conditions, and with the proper cross-analysis, to prove the major claims that this factor is an essential communicator of this modification at genes of interest. The authors need to tell their story in light of this seminal manuscript: Domcke, S., et al 2015. Competition between DNA methylation and transcription factors determines binding of NRF1. Nature, 528(7583), pp.575-579.

Reply

This reviewer also suggested that NRF1 ChIP-seq should be performed in the present study. We agree with this comment, and an additional NRF1 ChIP-seq analysis was performed with MCF-7 and ADR cells.

The comparison of NRF1 ChIP-seq peaks in ADR and MCF-7 cells revealed that more than 5000 new binding sites, in addition to those already occupied in MCF-7 cells, showed increased NRF1 binding after genotoxic adaptation (Fig. 7A, Supplementary Fig. S22 and Table S15). Newly bound NRF1 sites correlate with the ADR COGC-seq signal, and a high H3K27ac signal was also measured in these regions (Figs. 7B and C). A large fraction of these sites (55.4%) occurred at promoter regions in ADR cells, which suggested that genotoxicity-induced O-GlcNAc enhanced the transcriptional regulatory function of NRF1. We also performed WT-NRF1 (wild-type NRF-1) and AA-NRF1 (O-GlcNAc amino acid site mutant NRF1) ChIP-seq with stably transfected MCF-7 and ADR cells (Supplementary Fig. S22). Notably, even though the two types of recombinant NRF1 exhibited similar expression levels, AA-NRF1 showed significantly less chromatin binding than WT-NRF1 (Fig. 6I), which indicated that the abundance of chromatin-bound NRF1 is monitored by O-GlcNAc modification. The increase in NRF1 binding was also

accompanied by a significant increase in the expression of downstream genes (Fig. 7D). Our NRF1 ChIP-seq analysis with two cell models clearly confirmed that O-GlcNAc promotes this TF dynamic assembly with chromatin during the genotoxic stress response. Further O-GlcNAc inhibition and downregulation of NRF1 also suppressed these genes and protected against Adm stress in ADR cells (Supplementary Fig. S21). Therefore, O-GlcNAc modification can enhance the transcriptional activation function of NRF1 and protect cancer cells from genotoxic stress. We have added a related discussion and revised the text in the revised manuscript.

General comments:

The paper is hindered greatly by the incorrect grammar throughout. Before this is acceptable for publication in Nature Communications, this must be cleaned up to make it readable. Or, resubmit to Nature Miscommunications. This reviewer has found an error in almost every sentence, but I will not correct all of them.

Reply

We accepted this reviewer's comment and the English language in revised manuscript has been checked by *American Journal Experts*.

Specific remarks:

- 1. To improve the interpretability of "ADR", in the figures I recommend having the full definition of this term presented upfront, perhaps with a schematic illustration of the several months of induced Adriamycin resistance achieved by the authors. (and, how many months specifically? 3 months? 14 months?)**

Reply

We accept this reviewer's comment. The full definition of "ADR" was added in

the revised when it first appeared. The schematic illustration of the 8 months of induced Adm resistance was also added in revised Fig. 2D.

2. Line 61 – edit to “levels in a stress”...

Reply

We agree with this reviewer’s comment and revised certain text.

3. Line 71 – this sentence is incorrect grammar “Recently, chromatin O-GlcNAc modification is involved in gene expression”. It is missing a phrase such as “it has been discovered that chromatin O-GlcNAc...”

Reply

We agree with this reviewer’s comment and revised certain text.

4. Line 103 – delete the incorrectly placed word “in”

Reply

We agreed with this reviewer’s comment and revised certain text.

5. Figure 1a: It would be easier to appreciate the method if azide (N3) was in bold in the chemical structure.

Reply

We agree with this reviewer’s comment and revised Fig. 1A and 2D.

6. Figure 2c: If O-GlcNAz is indiscriminately distributed to all transcribed genes, this plot would be expected to mirror RNA-seq (or, better, proteome-wide data without an O-GlcNAz pull down). The authors need to repeat this volcano plot as a series of volcano plots, where each plot has the

gene from different RNA-seq Log₂FC bins (to see how much of this difference can be a simple reflection of expression differences). I might recommend binning each gene in the volcano plot into “Up in MCF7”, “Not differentially expressed” between MCF7 and ADR, and “Up in ADR”, using a threshold of ~Log₂FC of 3 for TPM RNA-seq values.

Reply

We agree with this reviewer’s suggestion and added figures in the supplementary files. We repeated the volcano plot in Fig. 2E to show a series of volcano plots.

The OCPs are divided into three bins according to FPKM fold change of the corresponding genes obtained by RNA-seq: “OCPs mRNA levels upregulated in ADR” ($\text{Log}_2 \text{ fold change (FC)} \geq 3$), “OCPs mRNA levels not differentially expressed between MCF-7 and ADR” ($3 > \text{Log}_2 \text{ FC} > -3$) and “OCPs mRNA levels upregulate in MCF-7” ($\text{Log}_2 \text{ FC} \leq -3$, Supplementary Fig. S7A). Invisible transcriptional differences were found in OCPs identified by LC-MS/MS. Furthermore, the OCPs identified by quantitative proteomics were divided into three bins according to the corresponding fold change in protein level (protein must be identified in six out of nine replicates of at least one group, $\text{Log}_2 \text{ FC} \leq -1$ or ≥ 1 , $p \text{ value} \leq 0.05$, $\text{FDR} \leq 0.05$) by whole-cell quantitative proteomics without O-GlcNAz enrichment: “OCP protein levels upregulated in ADR”, “OCP protein levels not differentially expressed between MCF-7 and ADR” and “OCP protein levels upregulated in MCF-7” (Supplementary Fig. S7B).

The vast majority of OCP quantities could not reflect the gene expression differences in the RNA-seq and whole-cell proteomics data without O-GlcNAz enrichment. This finding might have been obtained because only a part of the protein of each TF could undergo O-GlcNAc modification and interact with chromatin. Other parts of each TF, which do not undergo O-GlcNAc modification or interact with chromatin, could not be enriched using this chemical-reporter-based strategy. Our results indicated that O-GlcNAc modification regulates the chromatin binding of these TFs. These results also certified the specificity of this chemical-reporter-based OCP

enrichment.

- 7. Figure 3a: The ADR vs. MCF7 motif comparison is rather weak on a few fronts. First, the GGGnGGG motifs for ADR are not different essentially from the CCCnCCC motifs found in MCF7 – they are just the mirror image of one another. Presumably a P-value of enrichment for each motif is available for each cell line – why not, with each top ranking motif, share the motif enrichment across both cell lines?**

Reply

We agree with this reviewer's suggestion and revised Fig. 4A. MCF-7 and ADR-biased COGC-seq peaks were reanalyzed using motif discovery algorithms (Homer). Each top ranking motif with a p value across MCF-7 and ADR-biased COGC-seq peaks is shown. Because SP1 and KLF5 belong to the same TF family and bind to similar DNA motifs, the SP1 motifs are not essentially different from the KLF5 motifs found in MCF-7 and ADR-biased COGC-seq peaks. We have revised the text in the new manuscript.

- 8. Fig 4c: The legend has triangles “targeting less genes” twice.**

Reply

We agree with this reviewer's suggestion and revised the certain text.

- 9. Figure 3D/E – the authors need to perform GO enrichment analysis, perhaps using GREAT (<http://great.stanford.edu/public/html/>) or something similar, on their peaks (maybe on the overlapping peaks, and also on the non-overlapping peaks from each cell line). Something like that would be needed to justify the “leap” into genotoxic stress, over other possible gene pathway categories this mark might decorate.**

Reply

We agree with this reviewer's suggestion and performed additional GO enrichment analyses. Among the differential COGC-seq peaks between MCF-7 and ADR, 1572 O-GlcNAz-associated genes were uniquely found in MCF-7, whereas 1042 genes were exclusively observed in ADR. A GO analysis revealed that genes associated with ADR-biased peaks were more enriched in roles related to the stress response and DNA damage response than those of MCF-7-biased peaks (Fig. 4E).

10. Figure 6 – What does AA mean in “Flag-AA-NRF1”? Can you define this in the legend?

Reply

We agree with this reviewer's suggestion and added the definition for “AA-NRF1” in the revised manuscript. We also revised the legends of Fig. 6F.

REVIEWERS' COMMENTS

Reviewer #1 (Remarks to the Author):

The manuscript has been significantly improved from the last version, both in scientific content and English language readability. The authors have adequately addressed my concerns in terms of proteomic experiments, the actual role of O-GlcNAc modification as well as functional validation of NRF1. I think it is ready for publication.

Reviewer #4 (Remarks to the Author):

The authors have done a fantastic job revising this manuscript. While the paper isn't perfect, the improvements over the original submission is a substantial, and worth publishing without delay. I applaud the hard work and I hope the insights gained here will be quite useful for the field.